# How Different Substitution Positions of F, Cl Atoms in Benzene Ring of 5-Methylpyrimidine Pyridine Derivatives Affect the Inhibition Ability of EGFR^L858R/T790M/C797S^ Inhibitors: A Molecular Dynamics Simulation Study

**DOI:** 10.3390/molecules25040895

**Published:** 2020-02-18

**Authors:** Jingwen E, Ye Liu, Shanshan Guan, Zhijian Luo, Fei Han, Weiwei Han, Song Wang, Hao Zhang

**Affiliations:** 1Laboratory of Theoretical and Computational Chemistry, Institute of Theoretical Chemistry, Jilin University, Changchun 130023, China; ejw17@mails.jlu.edu.cn (J.E.); luozj17@mails.jlu.edu.cn (Z.L.); hanfei15@mails.jlu.edu.cn (F.H.); 2Key Laboratory for Molecular Enzymology and Engineering of Ministry of Education, School of Life Science, Jilin University, Changchun 130012, China; lye16@mails.jlu.edu.cn (Y.L.); weiweihan@jlu.edu.cn (W.H.); 3College of Food Engineering, Jilin Engineering Normal University, Changchun 130052, China; guanshanshan@jlenu.edu.cn; 4Key Laboratory of Molecular Nutrition at Universities of Jilin Province, Changchun 130052, China

**Keywords:** oncogenic EGFR, non-small-cell lung cancer, inhibitor, epidermal growth factor receptor, molecular dynamics simulation, MM-PBSA

## Abstract

Lung cancer is the most frequent cause of cancer-related deaths worldwide, and mutations in the kinase domain of the epidermal growth factor receptor (EGFR) are a common cause of non-small-cell lung cancers, which is a major subtype of lung cancers. Recently, a series of 5-methylpyrimidine-pyridinone derivatives have been designed and synthesized as novel selective inhibitors of EGFR and EGFR mutants. However, the binding-based inhibition mechanism has not yet been determined. In this study, we carried out molecular dynamic simulations and free-energy calculations for EGFR derivatives to fill this gap. Based on the investigation, the three factors that influence the inhibitory effect of inhibitors are as follows: (1) The substitution site of the Cl atom is the main factor influencing the activity through steric effect; (2) The secondary factors are repulsion between the F atom (present in the inhibitor) and Glu762, and the blocking effect of Lys745 on the phenyl ring of the inhibitor. (3) The two factors function synergistically to influence the inhibitory capacity of the inhibitor. The theoretical results of this study can provide further insights that will aid the design of oncogenic EGFR inhibitors with high selectivity.

## 1. Introduction

Cancer is a kind of disease characterized by abnormal cell proliferation and metastasis. It has the characteristics of high morbidity and mortality. It is one of the malignant diseases that threaten human health and lead to death. Some protein kinases’ overexpression or abnormal expression in cancer cells leads to abnormal signal transduction, which is involved in many physiological processes such as cell proliferation, differentiation, metabolism, apoptosis, etc. The pathogenesis of non-small cell lung cancer (NSCLC) is closely related to the mutant kinase activity in epidermal growth factor receptor (EGFR) [1,2], while abnormal signal transduction is closely related to the occurrence of cancer; it can accelerate cellular apoptosis [3], antagonize angiogenesis [4], and inhibit tumor metastasis [5] and tumor growth [6]. Therefore, it has become one of the most important research fields to design and develop kinase inhibitors that act on oncogenic EGFR.

EGFR is a transmembrane glycoprotein belonging to the tyrosine kinase receptor family and composed of 1186 residues, including three parts: extracellular receptor region, transmembrane structure, and intracellular tyrosine kinase region [7,8]. The structure and function of the intracellular domain of EGFR (from Gly696 to Ile1018) has been a major topic of research. The structure of the inhibitor bound to the intracellular domain of EGFR is presented in Figure 1. There are seven α-helices and seven β-sheets in the intracellular domain, and the inhibitor can be firmly locked in the ‘mouth-like’ structure and compete with the natural substrate for the ATP binding site. The groove in the ‘mouth’ consists of five functional regions, which are the P-loop (Ser720 to Gly724), α-loop (Arg748 to Ser752), Cα-helix (Pro753 to Ser768), hinge-region (Gln791 to Leu798), and DFG motif (Thr854 to Arg858).

The EGFR gene is located on the short arm of chromosome 7 and contains 28 exons [9], and the exons 19 and 21 of EGFR are the sites of hotspot mutations in lung adenocarcinomas [10]. At present, the targeting drugs for oncogenic EGFR are mainly divided into two categories according to their properties [11]. One is the monoclonal antibody [12,13], which directly acts on the extracellular receptor region. The other one can inhibit the catalytic activity by binding to the tyrosine kinase catalytic region of EGFR [14]. Both EGFR-targeted therapies can block the cell’s proliferative signal [15,16]. Since EGFR is prone to pathogenic mutations that could cause deactivating inhibitors, new inhibitors targeting EGFR mutants are constantly being developed [17,18,19]. Until now, the four generations of EGFR tyrosine kinase inhibitors (TKIs) have been approved and developed in all. Some EGFR TKIs that have been researched are listed in the following Table 1 with remarkable results. There are mainly first-generation EGFR inhibitors, such as gefitinib (Iressa) [20] and erlotinib (Taeceva) [21], which have proven to be a successful therapeutic strategy for the treatment of NSCLC [22]. At present, the second-generation EGFR inhibitors that have been listed and are mainly clinical the afatinib (Giotrif) [23], dacomitinib [24], and neratinib [25]. In order to improve the selectivity of pathological EGFR and wild-type EGFR, scientists have developed the third-generation inhibitors, such as the CO-1686 (rociletinib) and AZD9291 (osimertinib), which were approved for the treatment of locally advanced or metastatic EGFR^T790M^ mutation-positive NSCLC. This covalent binding can be hindered by mutating cysteine residues to serine (C797S mutation) [26]. For the above reasons, scientists began to study new small molecule inhibitors for the fourth-generation EGFR (EGFR^L858R/T790M/C797S^) [27]. At present, EAI045 is the first noncompetitive inhibitor designed for EGFR triple mutation [28]; however, it needs to be combined with an EGFR monoclonal antibody to effectively exert its inhibition effect. Another reversible small molecule inhibitor was called Brigatinib, which used ALK/EGFR dual-target combination therapy [29]. Recently, a series of 5-methylpyrimidine-pyridinone derivatives have been designed and synthesized at Jinan University, Jinan, China, as novel EGFR^L858R/T790M/C797S^ (EGFR^TM^)-selective inhibitors [30]. To ensure their high inhibitory activity, various molecular scaffolds with different biological effects have been employed. The experimental results reveal that these inhibitors exhibit a higher inhibition efficiency against EGFR mutants (EGFR^TM^) compared with wild-type EGFR (EGFR^WT^). Interestingly, different substitution positions of the fluorine atom and chlorine atom on the benzene ring of 5-methylpyrimidinone derivatives will result in great differences in inhibition ability for EGFR^TM^. The six inhibitors are 8r-B (IC_50_ = 27.5 ± 11.6 nM), 8r-A (IC_50_ > 1000 nM), 8p-B (IC_50_ = 37.1 ± 1.2 nM), 8p-A (IC_50_ = 207.0 ± 135.0 nM), 8q-B (IC_50_ = 88.6 ± 13.3 nM), and 8q-A (IC_50_ = 224.1 ± 6.7 nM), and the specific conformation is shown in Figure 2.

In view of the above experimental values, the inhibitory mechanism and structure–activity relationship of these inhibitors have not been studied at the molecular level; hence, a complete understanding of the inhibitory mechanism is lacking. The mutant model of EGFR^TM^ was built with EGFR^T790M^ protein (PDB code: 5XDK) as a template and optimized the EGFR^TM^ under a relaxation state. Six inhibitors (shown in Figure 2) were constructed by Gaussian 09 software. According to the experiment results, these inhibitors bound to in the ‘mouth’, which consisted of the P-loop, α-loop, Cα-helix, hinge-region, and DFG motif. Inhibitors were docked into EGFR^TM^ using by AutoDock vina software. 

In this study, the model EGFR^TM^ inhibitors were established and analyzed. These results establish the complexes model to understand the fourth-generation inhibitors’ EGFR^TM^ inhibitory mechanism and provide a basis for the rational design of reversible EGFR^TM^-based inhibitors with improved EGFR^TM^ potency and selectivity.

## 2. Results and Discussion

### 2.1. The Binding Conformations Analysis

All six inhibitors belong to reversible inhibitors which are located in the cavity of the hydrophobic environment, and the location of these competitive inhibitors is the reported ATP binding site [38,39,40,41,42,43], which is involved van der Waals interactions and electrostatic interactions. It is observed that there are nearly 20 amino acids in 3.5 Å around the inhibitors from Appendix A, which are hydrophobic amino acids—Leu718, Gly719, Val726, Ala743, Met766, Leu777, Leu788, Met790, Leu792, Met793, Pro794, Phe795, Gly796, Leu844—and hydrophilic amino acids—Lys745, Cys775, Gln791, Ser797; it can be seen that the inhibitors are bounded to a large hydrophobic cavity, which is defined as “mouth” and is the same as below. It is worth mentioning that when the F atom is at position 5 and the Cl atom is at position 6, namely 8r-B can form a double hydrogen bond with Met793 and a double hydrogen bond with Ser797, while when the F atom is at position 3 and the Cl atom is at position 2, 8r-A only forms a single hydrogen bond with Met793 (shown as Appendix A). When the F atom is at position 3, the Cl atom is at position 6; then, both Met790 and Lys745 form strong conjugation interaction with the benzene ring head of 8p-B, and the double hydrogen bond with Met793 also exists stably. When the F atom is at position 5 and the Cl atom is at position 2, 8p-A still forms a double-hydrogen bond with Met793, but the hydrophobic residue Gly796 is close to the hydrogen bond, which may cause interference (shown as Appendix A). When the F atom is at position 4 and the Cl atom is at position 6, the double-hydrogen bond of inhibitor 8q-B with Met793 and the conjugation of Lys745 with a benzene ring stabilize the whole inhibitor skeleton. When the F atom is at position 4 and the Cl atom is at position 2, Met793 does not form a hydrogen bond with inhibitor 8q-A, and the conjugation interaction of the benzene ring head is still formed by Lys745 (shown as Appendix A). The difference of the above docking positions can also be obtained from the comparison of the docking affinity in Table 2. Based on the above considerations, when the Cl atom is at position 6, the whole inhibitor will bond more firmly with the residues around the protein, i.e., 8r-B, 8p-B, and 8q-B. In order to explore the specific dynamic process of interaction, molecular dynamics simulation can explain the binding-based inhibitory effect well.

### 2.2. Molecular Dynamics Trajectory Stability and Flexibility Analysis

The root-mean-square deviation (RMSD) of the backbone atoms, with respect to the initial position, is usually regarded as an indicator of systematic stability. Firstly, the RMSD of the protein backbone were calculated to describe the conformational changes of the six systems. As shown in Appendix A, all six systems remained comparatively stable for at least 20 ns. Especially, the average RMSD value of Appendix A showed that the 8r-B-bound complex was the more stable trend than the other conformations co-factor bounds. It should be noted separately that the RMSD average values of the five main functional areas are shown in Figure 3a. It can be seen from the figure that the value of the 8r-B-bond complex is the lowest, and the protein is more firmly bound by the inhibitor. On the contrary, the value of the 8r-A-bond complex is the highest, which is also an indication of the instability of active hydrophobic cavity. Further, the time-dependent solvent-accessible surface area (SASA) can be able to quantify the area of enzymes exposed to the solvent. It is now a commonly accepted concept that the active site of the enzyme is located in the hydrophobic cavity of protein. Since the substrate is bound to the active site of the protein, the interaction between the substrate and the catalytic group will be stronger in the non-polar environment, while the average value of SASA for the active binding pocket (Figure 3b) showed that the SASA contribution of the 8r-B-bound complex was the lowest and that of the 8r-A-bound complex was the highest among the six systems, which indicated that the EGFR^TM^_8r-B complex could provide a decent hydrophobic environment for the substrate binding. All these suggested that the protein and substrate (8r-B) showed a mutual stabilizing effect for each other but 8r-A does not do this stabilizing effect.

Regional conformational fluctuations can be assessed by calculating the Cα root-mean-square fluctuation (RMSF), which demonstrates the deviation of each residue from the MD (molecular dynamics) trajectory. RMSF is one of the best methods of comparing the dynamics and profiles of all the residues of the protein Cα that are represented in Figure 4a; parts of the protein that are highly flexible will have a large RMSF value, while portions that are constrained will result in a low RMSF [44]. The RMSF of some of the residues present in the flexible functional area are represented in Figure 4b–e. Two phenomena can be observed in Figure 4; for all complexes, the RMSF value of the Leu979-Ile1018 region, which is at the C-terminal, was the highest. It is usual for the unconstrained and outer-sphere ending loop except for the residues from Leu979 to Ile1018, such as the P-loop (Ser720 to Glu724), α-loop (Arg748 to Ser752), Cα-helix (Pro753 to Ser768), hinge-region (Gln791 to Leu798), and the active-loop (DFG-motif, Thr854 to Pro877), which comprise the inhibitors’ binding functional regions and exhibited relatively flexible and significant in different complexes. It can be seen from the whole RMSF that the overall fluctuation of the 8r-B-bound protein skeleton is relatively low, indicating that the protein and inhibitor are strongly constrained. On the contrary, the overall fluctuation of the 8r-A-bound protein skeleton is relatively large, and it is specifically reflected in the functional regions mentioned above. In particular, the Cα-helix region had a significant RMSF value, indicating that it is strongly affected by substrate binding. The P-loop, α-loop, Cα-helix, and active-loop are all present at the phenyl terminal of the substrate in the form of a benzene ring with halogen substitutions, while the phenyl rings could show a fair degree of mobility and explore a relative wide range of conformations when the four regions exhibited relatively flexible properties. The difference in the RMSF values of the six complexes indicates that different substitution sites may affect these regions differently. The hinge-region was close to the opposite terminal of the substrate; residues Met790 and Ser797 of this region were mutated in the second and third-generation mutations. The alteration in the hinge-region seems to be associated with the selective inhibitory action of the inhibitors against different mutants. Since the present study was primarily concerned with the inhibitory activity of different inhibitors and was not of the different protein’s mutants, the intention to study the changes in the hinge-region will be researched in the future. As seen in Figure 4b–e, 8r-A had the largest RMSF value in all four regions, which indicates that 8r-A exerts a significant effect on the conformational changes of these regions. 

In order to monitor the six systems’ dynamics behaviors, the cross-correlations of their Cα atomic displacements could determine whether the six systems demonstrated similar dynamic behavior, which can reflect the detailed atomic dynamic state and were analyzed during the simulations [45]. Here, the linear correlations of each pair of Cα atoms were depicted by constructing covariance matrix maps based on the first two eigenvectors. The covariance matrix maps of the six complexes are illustrated in Appendix A, where the anti-harmonic and large-scale motions are highlighted by diagonalizing the matrix. The highly positive regions (blue) indicate a strong positive correlation in the motion of the residues, whereas the negative regions (red) are associated with strong negative correlation in the motion of the residues. In general, the motion lies in the normal range when the values fluctuate between −0.2 and 0.2. As expected in Appendix A, the most apparent correlated motions occurred in the region spanning from Arg986 to Val1010 in all the covariance matrix maps. Moreover, two regions with correlated motion are significant (as demonstrated by the square boxes in Appendix A). The first is the residue sequence from Thr710 to Val765, which contains the P-loop (Ser720 to Gly724) and the α-loop (Arg748 to Ser752). The second is the residue sequence from Cys753 to Trp880, containing the Cα-helix (Pro753 to Ser768), the hinge-region (from Gln791 to Leu798), and the active-loop (from Thr854 to Pro877). These regions were corresponded with those observed in the RMSF and Principal Component Analysis (PCA) analysis. In addition, inhibitor 8r-A leads to the strongest centralized self-correlated motions, whereas inhibitor 8r-B leads to the weakest ones. This might be attributed to the significant restriction of inhibitor 8r-B binding to the enzyme, and this conclusion was consistent with RMSF values.

To further confirm the above results, the defined secondary structure of protein (DSSP) analysis was performed, and the results are shown in Figure 5. The small changes exhibited by the activation segments of the 8r-A-bound protein and Cα-helix region are more significant than those exhibited by the activation segments of other complexes. The Cα-helix region has occurred looseness at the end of the simulation from the Figure 5b; in other words, the structure of the α-helix, 5-helix, and 3-helix had a mutual transformation. This secondary structural change could be confirmed by RMSF analysis, and other functional areas binding with inhibitors are loop areas, so there is no apparent difference in the secondary structure (shown as Appendix A). The specific functional areas change could made by the porcupine plot for the PCA.

### 2.3. The Porcupine Plot for the Principal Component Analysis of the Six Complexes

PCA analysis represents the trajectory of motion of each part of the protein during the entire simulation process. Figure 6 and Figure 7 represent the motion of the carbon framework of the free protein and the six complexes. It can be observed that the region had large motion spans from Leu979 to Ile1018 in all complexes. In addition, when no inhibitor interacted with the protein (apo protein), the P-loop and the α-loop of the protein move rapidly toward the direction of water, which resulted in the expansion of the whole protein, as represented in Figure 6. When the 8r-B inhibitor was bounded to the protein, the loop region adjacent to the substrate had an apparent tendency to reduce the width of the substrate binding pocket (i.e., the ‘mouth-like’ pocket), especially on the phenyl ring ending in the EGFR^TM^_8r-B complex (represented as Figure 7a). The motion would increase the difficulty in dissociating the substrate from the protein. In contrast, when the inhibitor 8r-A was bounded to the protein, it can be found that the direction of movement of the loop regions around the substrate tended to increase the width of the pocket and eased the dissociation of the substrate (as shown in Figure 7b). The trend of P-loop motion in other EGFR^TM^-inhibitor complexes is insignificant compared with 8r-B and 8r-A complexes (Figure 7c–f). These results were corresponded with those deduced from RMSF analysis.

### 2.4. FEL and Clustering of the Sampling

Then, the lowest energy conformation was selected in the free energy surface from the most abundant clusters as a part of stable low-energy conformation for the analysis object. Free-energy landscape (FEL) was used to obtain several low-energy conformations, which are the most suitable analytical subjects according to RMSD-equalized trajectory. It can be observed that the P-loop, α-loop, and activity-loop regions had relatively large conformational changes. One double hydrogen bond can be seen between Met793 and the inhibitors. In all systems, the existence rate of this hydrogen bond is more than 60% in all the molecular dynamics simulations (as listed in Table 3). The hydrogen bond was also observed in the docking result and the crystal.

From the equalized trajectory, cluster analysis was used to get insight into possible conformational states [46]. Then, the cluster analysis method was applied to obtain a series of stable conformations from the equilibrium trajectory, shown as Appendix A. Cluster analysis results show that there are four clusters in the EGFR^TM^_8r-B stable locus, seven clusters in the EGFR^TM^_8r-A stable locus of the complex, five clusters in the EGFR^TM^_8p-B stable locus of the complex, five clusters in the EGFR^TM^_8p-A stable locus, eight clusters in the EGFR^TM^_8q-B stable locus of the complex, and six clusters in the EGFR^TM^_8q-A stable locus of the complex. In addition, some clusters whose proportion is less than 2% are ignored because of their small proportion in the analysis. 

After comparing the conformations of each cluster, there were some difference in the loop region (Asp974-Ile1018), but this flexibility belongs to the N-terminal of the protein and was related to the structure of the protein itself. This can be verified with RMSF results. As represented in Figure 8, showing FEL combined with cluster analysis, two representative structures for the six complexes were sampled with the corresponding structures at 41.59 and 93.8 ns in EGFR^TM^_8r-B; 38.12 and 99.8 ns in EGFR^TM^_8r-A; 83.7 and 88.36 ns in EGFR^TM^_8p-B; 60.12 and 70.66 ns in EGFR^TM^_8p-A; 38.12 and 76.49 ns in EGFR^TM^_8q-B; and 46.67 and 62.11 ns in EGFR^TM^_8q-A. For PC1, snapshots captured at 41.59, 99.8, 83.7, 60.12, 76.49, and 62.11 ns conformations for EGFR^TM^_8r-B, EGFR^TM^_8r-A, EGFR^TM^_8p-B, EGFR^TM^_8p-A, EGFR^TM^_8q-B, and EGFR^TM^_8q-A, respectively, which were selected as stable conformations to analyze the interactions between the protein and the substrate. The six stable conformations were named EGFR^TM^_8r-B, EGFR^TM^_8r-A, EGFR^TM^_8p-B, EGFR^TM^_8p-A, EGFR^TM^_8q-B, and EGFR^TM^_8q-A. The substrates were still bound close to their initial positions in the selected conformations.

### 2.5. Conformational Analysis of Factors Influencing Inhibition Ability

In this section, we will discuss the conformational changes mentioned above at the residue level. In Figure 9, we observed that the ligand is surrounded by a ‘mouth-like’ pocket including approximately 20 residues. Salt bridges can be observed on both sides of the ‘mouth-like’ pocket [47,48] (yellow region in Figure 9). The salt bridges might lock the pocket, block the dissociation of the inhibitor, and affect its ability to inhibit. On the left side, positive Arg999 and negative Glu804 lock the piperazine end of the substrate. Although the right (-phenyl) side has been primarily discussed in this article, a quadrilateral can be seen, which was involved with the residues, Lys745, Glu762, Asp855, and Arg858, (represented by the enlarged square frame in Figure 9). Among these residues, Asp855 and Arg858 were located in the loop region (active-loop), which imparts a significant flexibility to the entire simulation process. Therefore, Lys745 (on the β-sheet close to the α-loop) and Glu762 (on the Cα-helix) may possess greater stability and exert a significant effect on activity. The effect of salt bridges will be discussed in greater detail in the following sections. 

In the experiment, the inhibitors exhibit variable inhibition ability, which depended on the substitution sites of halogen atoms. The different substitution positions of F and Cl atoms on the benzene ring of the inhibitor result in different inhibition ability, which is the focus of this study. The halogen atoms are all hydrophobic atoms, and the benzene ring head is also in a strong hydrophobic cavity, which was composed of Val726, Ala743, Met766, Ile788, Met790, and Met793. It can be seen that when the F and Cl atoms are embedded in the hydrophobic cavity, the benzene ring of the whole inhibitor will be the most stable, while when the F and Cl atoms are exposed outside the “mouth”, the benzene ring of the whole inhibitor will be weakly bounded to the hydrophobic cavity. This conclusion is mutually confirmed by the lowest IC_50_ value of 8r-B and the highest IC_50_ value of 8r-A. However, when the F and Cl protons are replaced on the benzene ring separately, the inhibition ability also different, which is the focus of our next exploration.

In addition to the strong stabilizing effect of the hydrophobic cavity on the head of the benzene ring, from the IC_50_ value of the experimental data, it is found that only one halogen atom of the F atom and the Cl atom in the hydrophobic cavity will also cause the difference of the inhibition ability, so the atomic charge distribution of different inhibitors was calculated by different methods of Mulliken, Nature Bond Orbital (NBO), and Restrained ElectroStatic Potential (RESP), as shown in Table 4 below. Interestingly, the negative charge of the F atom is strong, while the negative charge of the Cl atom is not high and approximately 0; that is, the Cl atom carries no charge in all inhibitors. Moreover, the charge difference of atoms with different ligands in RESP methods showed that the F atom had the highest negative charge and the difference of charge between the six inhibitors is insignificant (as represented in Figure 10). It appears that the uncharged Cl atom cannot influence the salt bridge, since it does not interact with the protein. Nevertheless, the experimental results reveal that the inhibition ability is determined by the position of the Cl atom. When the Cl atom is located at position 6, the inhibition ability is higher (IC_50_ value is less) than when the Cl atom is located at the position 2. Thus, we contemplated whether a large volume of Cl atom could block the dissociation of the inhibitor. As is apparent from conformational comparison, the Cl atom is surrounded by Val726, Lys745, and Met790 when it is located at position 6, while it is exposed to the solvent when positioned at 2 (as depicted in Figure 11). Understandably, in the latter case, it is easier for the inhibitor to interact with the solvent due to less steric hindrance. For competitive inhibitors, an ease of substrate dissociation is closely related to the inhibitory activity. When the Cl atom is at position 6—that is, when the Cl atom is embedded in the hydrophobic cavity—it is always stronger than the inhibition ability when the Cl atom is exposed outside the “mouth”. Table 5 lists the van der Waals radius of the main atoms wrapped in the benzene ring head, which were corrected by Pauling [49]. The Cl atom plays a major role in stabilizing the head of the benzene ring, which can be explained by the maximum van der Waals radius of the Cl atom in the table.

In addition to the location of the Cl atom, the location of the F atom significantly influenced inhibition ability, although it is not the determining factor. Why does the substitution of a strongly negative F atom at positions 3, 4, and 5 of the benzene ring cause different inhibition ability? It can be seen from Figure 9 that the benzene ring head is blocked by Lys745 and the positive polarity is mainly concentrated on the N atom, so the measurement of the distance between the F atom and N atom of Lys745 is shown in Table 6. The results show that the F atom will strongly attract Lys745 with positive charge; the distance between F atom and N proton of Lys745 is almost the same in different systems, and the two are almost kept at the same level. The position of the benzene ring head in the hydrophobic cavity will change slightly. At the same time, the electronegativity of the F atom will affect the negative Glu762 of the four-salt bridge system, which is mutually exclusive. The measurement of the distance between the concentrated positively charged N atom in Lys745 and the concentrated negatively charged O atom in Glu762, as shown in Table 6. It can be seen from the table that when the F atom is at position 3, i.e., 8p-B, 8r-A, because of the attraction of Lys745, the F atom will move to the upper part of the hydrophobic cavity, so that the repulsion with Glu762 will be very large, resulting in Glu762 moving to the outside of the “mouth”, and the hydrogen bond of Met766, Ile777, a hydrophobic amino acid on the same α-helix as Glu762, will break, resulting in the instability of the hydrophobic cavity (the hydrogen bond data of Met766-Glu762-Ile777 in different systems are shown in Table 6). In the same way, when the F atom is at position 5, the N atom is the farthest away from Lys745, and the damage to the salt bridge is the weakest. In this way, the damage of the hydrophobic cavity and the α-helix is the weakest, and the result can be consistent with the IC_50_ of 8r-B, 8p-A. When the F atom is at position 4, the head of the benzene ring of the inhibitor hardly moves, so the conclusion can also be confirmed by IC_50_ of 8q-B and 8q-A. Initially, conformational comparison was used to indicate that when the F atom occupies various substitution sites, the repulsion between the negatively charged F atom and Glu762 can induce a significant conformational change in the latter. When the F atom is at position 3 (8r-A), Glu762 experiences a strong repulsion away from the substrate (as shown in Figure 12), which can alter the conformation of the entire Cα-helix. This was corresponded to the RMSF result, in which the 8r-A has the largest RMSF value on the Cα-helix. The statistical average values of the distance between the negatively charged atoms (OE2) on Glu762 and the positively charged atom (NZ) on Lys745 are listed in Table 5. The location of the F atom at position 3 (the site nearest to Glu762) compared with 7.15 nm (8r-A) and 6.53 nm (8p-B), which are the longest distances listed, as observed from Table 5; this indicates significant disruption of the salt bridge between Glu762 and Lys745. In contrast, the values of 5.52 nm (8r-B) and 5.75 nm (8p-A) corresponded to the F atom at position 5 (the site farthest from Glu762). In addition, the F atom can attract Glu 762 by attracting positive Lys745, which could bring the Cα-helix and the β3-sheet in close proximity to each other. Table 5 enlists the distance between the negatively charged F atom and the positively charged N atom in Lys745 during molecular dynamics simulation. It can be observed from Table 5 that the average distance values of the six systems are similar. Figure 12 represents the last frame of dynamics simulation. Lys745 is seen to always be in front of the F atom, irrespective of the substitution site of the F atom. The position of the phenyl ring can be altered when the F atom is located at a different site. When the F atom is at position 3, the substrate moved inward and increased its coverage by Lys745; this kind of inhibitor exerts greater steric hindrance and thus has a stronger inhibitory effect.

The N atoms of Lys745 and Arg858 are positively charged, but the O atoms of Glu762 and Asp855 are negatively charged; the four charged atoms can induce corresponding charges on the residues. The opposite electric charges’ attraction formed a strong salt bridge system. The four salt bridges are between residues spanning Lys745–Glu762, Glu762–Arg858, Asp855–Arg858, and Lys745–Asp855. As illustrated in Figure 12, in the EGFR^TM^_8r-B system, the lengths of the salt bridges are 4.7, 5.9, 4.0, and 5.8, respectively. In contrast, for the EGFR^TM^_8r-A system, they are 10.6, 12.5, 8.6, and 9.0, respectively; the distance of each salt bridge increased to twice that in the EGFR^TM^_8r-B system. Further, we observed that the salt bridge system had been damaged; excluding the salt bridge between Glu762/Lys745, all three salt bridges—between Glu762/Arg858, Asp855/Arg858, and Lys745/Asp855 in the EGFR^TM^_8r-A complex—were significantly damaged, as Glu762 was repelled from Arg858 by the F atom. In summary, repulsion disrupts the salt bridges between Glu762 and Lys745 and between Arg858 and Asp855 to unlock the substrate binding pocket. This explains why 8r-A possesses the lowest inhibitory activity among all six inhibitors. 8p-B influenced by the effect of the Cl atom; possibly, Glu762 is significantly distant from the substrate, although it is not as distant as in 8r-A. In this system, the salt bridge between Lys745 and Glu762 is not entirely disrupted. Thus, the repulsive force exerted by the F-atom in 8p-B is much less than that in 8r-A. Further evidence on the substitution sites of halogen atoms in salt bridges in different inhibitors can be obtained from binding free energy analysis.

To investigate the interaction between the substrate and the active site, bottleneck residues were obtained by analyzing the molecular dynamics simulation trajectory through Channel Extraction and Visualization (CHEXVIS) [50,51,52], which revealed the structural details of the tunnel [53,54]. CHEXVIS was used to analyze 2500 snapshots from 100 ns molecular dynamics simulation of EGFR^TM^. Figure 13 illustrates the width, length, and bottleneck distance of the tunnel in EGFR^TM^_8r-B, EGFR^TM^_8r-A, EGFR^TM^_8p-B, EGFR^TM^_8p-A, EGFR^TM^_8q-B, and EGFR^TM^_8q-A complexes. The width represents the maximum movable range for inhibitors, which are surrounded by residues. The bottleneck can indicate the difficulty of dissociation of inhibitors from protein pores. A smaller bottleneck indicates a greater ease of the proteins in binding the inhibitors. As observed in Figure 13, both the width and bottleneck are related to inhibitory ability, although they do not strictly correspond to the order determined in the experiment. The three complexes on the left in Figure 13a,c,e possessed relatively smaller width and bottleneck and greater inhibition ability in the experiment than those on the right. In particularly, the EGFR^TM^_8r-A possesses the largest bottleneck, which might be related to a significant disruption of the salt bridge. The result indicates that the inhibitor does not dissociate easily from the substrate binding pocket, and therefore, it has stronger inhibition ability when the tunnel and its bottleneck become narrower.

From the above results, it can be concluded that there are three factors that might simultaneously influence the inhibition ability of inhibitors against EGFR^TM^. They are listed as follows: steric hindrance by the Cl atom, repulsion between the F atom and Glu762, and the position of the substitution site for the F atom, which can result in varying degrees of steric hindrance of the benzene ring by Lys745, via the interaction between the F atom and Lys745. Based on these factors, the order of inhibition by the six inhibitors mentioned in the previous sections was explained. 

Initially, the steric hindrance of the Cl atom is the main factor influencing the inhibitory activity. Thus, all three inhibitors in which the Cl atom is located at position 6 possessed higher inhibition activities. Among these inhibitors, 8r-B shows least repulsion, as its F atom is the least distant from Glu762. Moreover, the substrate faces the greatest hindrance from Lys745 in the EGFR^TM^_8p-B complex. Therefore, the two inhibitors have similar and the most significant inhibitory effects. In the EGFR^TM^_8q-B complex, Lys745 shows repulsive interaction and blocking ability, which is medial among the three inhibitors and explains the lower inhibitory capacity of 8q-B compared with 8r-A and 8p-A.

The Cl atom is located at position 2 in the other three inhibitors. Salt bridges between Lys745/Glu762 and Glu762/Arg858 are disrupted by repulsion in the EGFR^TM^_8r-A complex. The inhibitor 8r-A is the only one among the six inhibitors that can strongly disrupt the salt bridges. Thus, it possesses the lowest inhibition activity. Further, for 8p-A and 8q-A, the repulsion between the F atom and Glu762 is less significant in the former, while Lys745-induced blocking is more significant in the latter. The effect of these two factors is similar; hence, the inhibition activity of the two inhibitors is also similar.

### 2.6. Binding Energy Calculations

Molecular mechanics Poisson-Boltzmann surface area (MM-PBSA) is an efficient tool for analyzing protein–ligand interactions and confirming the observed results from the conformational comparison [55,56,57].

The total binding free energies of the six inhibitors of EGFR^TM^ are listed in Table 7. According to the table, the binding energy (ΔG_bind_), electrostatic interactions (ΔG_elec_), and van der Waals interactions (ΔG_vdW_) of the 8r-B complex are lower than those of the other inhibitors, indicating that 8r-B possesses the strongest competitiveness of all six inhibitors. It further was consistent with the results of the experiment.

The decomposed binding free energy of all the residues surrounding the phenyl terminal of the inhibitor is listed in Table 8 (their positions are denoted in Figure 14), which provides more detailed information on the interaction of each ligand and each residue. Some results can be deduced from the table. Firstly, the residues around the F atom possess the relatively lowest values of binding free energies, although they change when the F atom is located at a different site. Particularly, Lys745 possesses the largest value of ΔG_MM_ and ΔG_bind_ (greater than 20 kJ mol^−1^), irrespective of the site at which the F atom is located. This result confirms the strong interaction between the F atom and Lys745. Secondly, the order of binding free energy of Glu762 is consistent with the previously mentioned order of its distances from Lys745. When the F atom is located at the position 3 (8r-A and 8p-B), we observe the highest positive values of ΔG_MM_, denoting the greatest repulsive interactions. Nevertheless, significant negative values of 8r-B and 8r-A, as seen in Table 8, indicate that attractive interactions occur when the F atom is located at position 5. In the last two protein–inhibitor complexes, where the F atom is located at position 4 (8q-B and 8q-A), the smaller positive value of ΔG_MM_, compared to those in the 8r-A and 8p-B systems, indicate feeble repulsive interaction between Glu762 and the substrate.

## 3. Conclusions

Recently, a series of 5-methylpyrimidine-pyridinone derivatives that belonged to the fourth-generation competitive inhibitors of EGFR^TM^ were developed. However, different substitution positions of F and Cl atoms in the benzene ring lead to significant differences in inhibition. Therefore, this study analyzed the effect of the halogen atom substitution position on the affinity and selectivity for EGFR^TM^ protein. The binding conformations analysis indicated that all six competitive inhibitors located in the hydrophobic cavity stably, and the conclusion of difference affinity was consistent with the experimental IC_50_ value. The six complexes were investigated through MD simulation and binding free energy calculation. Through the RMSD, RMSF, SASA, and DSSP analysis of the whole trajectory and the key functional areas, the conclusion can be obtained: the small movement of the inhibitor and the change of hydrophobic cavity conformational could interact with each other, especially the 8r-B and 8r-A complexes. The cross-correlation and PCA analysis showed that when the inhibitor (8r-B) was embedded in hydrophobic pocket called the “mouth”, which was composed of five functional areas—the P-loop, α-loop, Cα-helix, hinge-region, and active-loop—the conformational change of 8r-B complexes will be very small, so the inhibitor will exist stably in the hydrophobic cavity and occupy the position of ATP relatively firmly. On the contrary, when the halogen atoms in 8r-A were exposed outside the hydrophobic cavity, the functional region of the protein was very loose, which resulted in the poor binding. The FEL and clustering of the sampling were analyzed for six complexes. Moreover, the charge distribution analysis and atomic charge calculation methods of Mulliken, NBO, and RESP displayed that the structure–activity relationship can be deduced by three factors. The substitution site of the Cl atom is the main factor affecting the activity through steric effect. The repulsion interaction between the F atom and Glu762 and blocking of the inhibitory phenyl ring by Lys745 are both secondary factors. Lastly, the secondary factors function synergistically influence the activity of the inhibitor. Thus, all the results reveal the mechanisms and the research can contribute and guide the design of competitive drugs for EGFR^TM^ in the future.

## 4. Materials and Methods

### 4.1. Initial Structure Preparation

Considering the structural similarity between EGFR^M790^ and EGFR^TM^ protein, the protein structure in 5XDK.pdb was selected; then, the missing residues were added by the SWISS-MODEL web-server and all water molecules and inhibitors were deleted from the Program Database (PDB) file, and finally, the first mutation L858R and third mutation C797S were created by Discovery Studio 4.0 software [58]. Considering the relaxation state of the oncogenic protein, the molecular dynamics simulation in GROMACS 4.5.2 was taken for the constructed protein structure [59] to make the protein as stable and relaxed as possible. The backbone atoms’ RMSDs value is shown in Appendix A, and the value is maintained at about 0.3 nm, indicating that the system has reached equilibrium. The comparison between the EGFR^L858R/T790M^ apo protein in 5EDP.pdb and the optimized protein (called “EGFR^TM^” in this study) is shown in Appendix A. The similarity of two constructed structures’ secondary structure is more than 80%, and the key functional areas’ structural similarity is more than 99% (shown as Appendix A); therefore, the final structure of the apo protein molecular dynamics simulation could be selected as the initial model for the next simulation. The structures of the six ligands (8r-B, 8r-A, 8p-B, 8p-A, 8q-B, and 8q-A) were fully optimized at the HF/6-31G(d) level using the Gaussian 09 program [60]. All structures were characterized as minima due to the absence of imaginary frequencies. 

### 4.2. Molecular Docking Calculations

The AutoDock Tools [61] were used for the setup of docking runs, and the molecular docking was simulated by AutoDock Vina software [62,63,64]. AutoDock Vina is a new molecular docking program that uses an elaborate optimization algorithm and method to complete the docking as well as the scoring function [63]. Since want to validate whether this complex is suitable for this software, the pre-processing of some of the PDB data was necessary, despite rectifications and updates from the organizers. Herein, the original inhibitor CO-1686 (*N*-[3-[[2-[[4-(4-ethanoylpiperazin-1-yl)-2-methoxy-phenyl]amino]-5-(trifluoromethyl)pyrimidin-4-yl]amino]phenyl]prop-2-enamide) in 5XDK.pdb was selected as the model for redocking analysis, so we adjusted the box to the center of the primary and secondary binding sites for the semi-flexible docking of the molecules. The size of the docking box chosen was approximately 12 × 14 × 18 points in the x, y, and z-axis directions; it was built with a gird spacing of 1 Å for CO-1686, the center of the grid was regarded as the center of mass, and the exhaustiveness was 20. The resulting box was used here as a reference for method validation. The pose of the ligand CO-1686 obtained from the docking resembled the one determined by X-ray crystallography, and the RMSD (root-mean-square deviation) of heavy atoms between the optimized pose obtained in our docking calculation and the one determined by X-ray crystallography was only 2.001 Å, demonstrating the high reliability of our docking scenario. The double hydrogen bond formed by Met790 and inhibitor was well reproduced in the result in Appendix A, the difference between the crystal ligand and the redocking model lay in the tail end of the inhibitor piperazine ring, which was exposed to the outside of the protein and entered the polar solvent: this was the reason for the large fluctuation. In Appendix A, the residues that involved van der Waals interactions and electrostatic interactions are the same, except for the ethanoylpiperazin of ligand, while other atoms reasonably presented high coincidence. Subsequently, the same docking strategy was used to elucidate the interactions between six inhibitors and the protein. Since the six inhibitors studied in this project are competitive inhibitors of ATP, the rectangular box is set at the ATP binding site, and all inhibitors are wrapped in it. Moreover, the docking method could reproduce the binding pose of the co-crystallized ligands in the used crystal structure. The dimensional grids were set as 14 × 16 × 12 points with a gird spacing of 1 Å for six inhibitors, and the exhaustiveness was 20. Appendix A shows that all six competitive inhibitors were successfully docked into the active site of the protein in a similar binding mode. The grid detail used in the docking simulations is shown in Appendix A. Finally, the six inhibitors were docked to the protein in turn, and the corresponding energy evaluations were also generated. The most frequently occurring and stable docking conformations were selected by docking with EGFR^TM^.

### 4.3. Molecular Dynamics (MD) Simulations

Conventional MD simulation was performed on the initial complexes using the GROMACS 4.5.2 software package [59]. The AMBER99SB-ILDN force field [65] was applied to build the parameter files of the protein, while the corresponding topology files of the ligands were generated using the Amber 14.0 program [66], with the charge distribution calculated by the AMBER force field “leap.gaff” [67]—a generalized amber force field. All complex systems were subjected to MD simulation in a periodic boundary condition of 10 Å with the Simple Point Charge (SPC) water model [68]. To keep each system at an electrically neutral charge state, seven sodium ions were randomly added to replace the water molecules. Prior to the MD simulation, the steepest-descent method was employed to minimize energy. The energy minimization structure was allowed to reach the initial equilibrium structure. Subsequently, the system was maintained in a stable environment (300 K, 1 bar) using 500 ps NVT (Berendsen temperature coupling with constant particle number, volume, and temperature) and 500 ps NPT (Parrinello–Rahman pressure coupling with constant particle number, pressure, and temperature). The coupling constants for temperature and pressure were set to 0.1 ps and 2.0 ps, respectively. The particle mesh Ewald algorithm with an interpolation order of 4 Å and grid spacing of 1.6 Å was used to describe long-distance electrostatic interactions, and van der Waals interaction was calculated with the cut-off value of 14 Å. All key lengths were constrained using the Linear Constraint Solver (LINCS) algorithm. After stabilization of the thermodynamic properties, the subsystem was simulated with 100 ns, a time step of 2 fs, and the coordinates of all models were saved every 10 ps [69,70,71].

### 4.4. Cross-Correlation Analysis.

Cross-correlation analysis was performed using Bio3D version 3.5.3 [72,73] to assess the extent to which the atomic fluctuations/displacement of a system are/is correlated with one another by examining the magnitude of all pairwise cross-correlation coefficients [74]. The Bio3D Dynamic Cross Correlation Map (DCCM) function returns a matrix of all atom-wise cross-correlations whose elements may be displayed in graphical representation, which is frequently termed a dynamical cross-correlation map or DCCM. In this map, the coordinate axis scale corresponds to the atomic number. The red region indicates that the more positive the matrix element value, the more positive the correlation is between the two atomic motion modes; the white region that is in agreement with the matrix element value is 0. More blue regions signify that the matrix element value is more negative, indicating that the corresponding two atomic motion modes tend to be negatively correlated. The diagonal of the matrix is the variance, which must be greater than zero, and so it is white or reddish.

### 4.5. Principal Component Analysis and Free Energy Landscape

Principal component analysis (PCA) can provide a highly detailed image of large-scale motions and the correlated movements of macromolecules in biological systems. PCA extracts correlate fluctuations from MD trajectories by reducing dimensionality using the covariance matrix of all conformations with respect to that of the average structure [75]. PCA provides orthogonal eigenvectors and their corresponding eigenvalues, based on the diagonalization of the covariance matrix [76,77]. The elements Cij in the matrix were defined as:(1)$$C_{ij}=\frac{<\Delta ri\cdot \Delta rj>}{{\left(<\Delta ri^{2}>\cdot <\Delta rj^{2}>\right)}^{1/2}}$$
where Δ*ri* (Δ*rj*) is the displacement vector corresponding to the *i*-th (*j*-th) atom (*i,j* = 1,2, …, 3N, where N is the number of atoms) of the systems, and ‹ › indicates an ensemble average. The eigenvectors of the matrix represent the directions of the concerted motions. In general, the first few principal components (PCs) describe the most significant slow modes related to the functional motions of a biomolecular system. In this study, PCA was performed using GROMACS 4.5.2 to investigate and compare the modes of motion of the six systems. The projections of the original structures are represented as plots of the cross-correlation map.

Free-energy landscape (FEL) can provide insights into the dynamic processes that occur in a biological system. In FEL, the free energy minima typically represent the conformational ensemble in a stable state, whereas the free-energy barriers denote transient states. The FEL is constructed on- the basis of the PCA data. FEL can be expressed as:(2)ΔG(X)=−KBTlnP(X)
where KB is the Boltzmann constant, T is the temperature of the simulation, and *P(X)* is the probability distribution along the reaction coordinate *X*. In our study, we calculated the FEL to identify the dominant conformational states with relatively low energies.

### 4.6. Cluster Analysis

Clustering of the structures of a trajectory can be accomplished using disparate methods (algorithms) and different criteria to judge structure similarity [78]. Here, according to the quality threshold-like (qt) method, RMSD-based clustering was performed to classify the similar structures in six trajectories of EGFR^TM^-inhibitors into the distinct group. The RMSD cut-off was set to 2.0 Å for each trajectory. After clustering, the proportion of each cluster was counted, and the similar conformation in trajectories of EGFR^TM^-inhibitors was split into the same cluster. The representative structures extracted based on the cluster analysis were used for further ensemble docking.

### 4.7. The Charge Distribution Analysis

Then, the charge distribution analysis and atomic charge of optimized inhibitors were calculated. We used three related calculation methods, which were the Mulliken charge [79] and Nature bond orbital (NBO) charge [80] calculated by Gaussian 09, and Restrained ElectroStatic Potential (RESP) charge [81] calculated by Multiwfn software [82]. 

### 4.8. Binding Free Energy Calculation

The MM-PBSA method was used to predict and evaluate the binding free energies and relative stabilities of different biomolecular structures; this method was also used to estimate the energy contribution of each residue to the binding energy [55,56,83,84,85]. In our study, we employed the g_mmpbsa tool for calculation. A total of 200 snapshots were chosen uniformly from the last 20 ns of the MD trajectory. The total binding energy (ΔG _bind_) was computed by the following equation:ΔG_bind_ = G_complex_ − (G_protein_ + G_ligand_)(3)
where ΔG bind represents the binding free energy between the protein and the ligand, the G complex represents the total free energy of the protein–ligand complex, and the G protein and G ligand are total free energies of the isolated protein and ligand in solvent, respectively. The binding energy is expressed as the combination of enthalpy and entropy terms:ΔG_bind_ = <E_MM_> − TS + <G_sol_>(4)
E_MM_ =E _bonded_ + E_nonbonded_(5)
E_bonded_ = E_bond_ + E_angle_ + E_torsion_(6)
E_nonbonded_ = E_elec_ + E_vdW_(7)
where <E_MM_> denotes the molecular mechanics energy of the molecule expressed as the sum of the internal energy of the molecule and the electrostatic and van der Waals energies; and the bonding energy (E_bonded_) includes angle, bonding, and twist energy. The nonbonded interactions (E_nonbonded_) include electrostatic (E_elec_) and van der Waals (E_vdW_) interactions and were modeled using the Coulomb and Lennard–Jones potential functions, respectively. The term ‘−TS’, represents the change in conformational entropy, and <G_sol_> is the free energy of solvation.

## Figures and Tables

**Figure 1 molecules-25-00895-f001:**
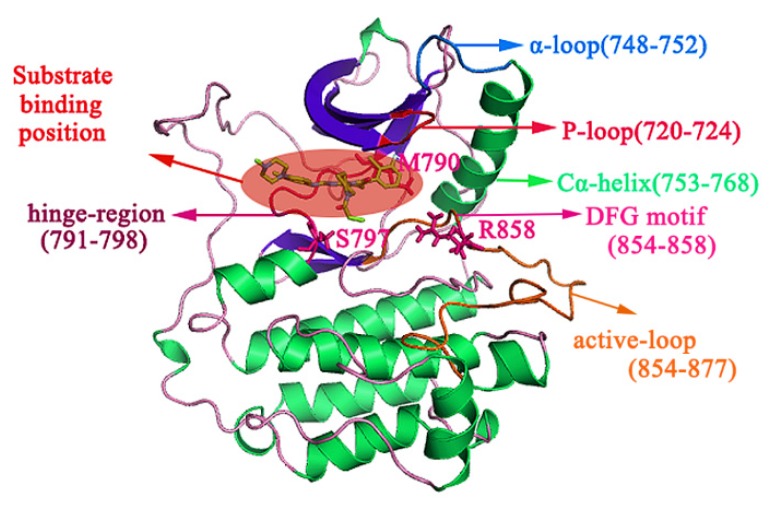
Cartoon representation of protein bound to substrate. The α-helices are marked in green and the β-sheets are marked in indigo. Residues S720 to G724 represent the P-loop (crimson); R748 to S752 represent the α-loop (blue); P753 to S768 represent the Cα-helix (green); Q791 to L798 represent the hinge-region (purple). The oncogenic mutant residues, M790, S797, and R858 are represented as pink stick structures, and the substrate binding pocket is represented as a red circle.

**Figure 2 molecules-25-00895-f002:**
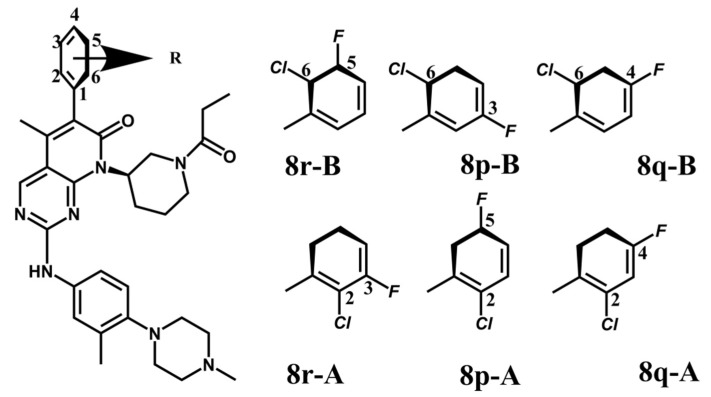
Structures of the inhibitors (generated using ChemDraw 18.0). As shown, position 5 of the benzene ring is substituted with F and position 6 is substituted with Cl, in 8r-B; position 3 is substituted with F and position 6 is substituted with Cl, in 8p-B; position 4 is substituted with F and position 6 is substituted with Cl, in 8q-B; position 3 is substituted with F and position 2 is substituted with Cl, in 8r-A; position 5 is substituted with F and position 2 is substituted with Cl, in 8p-A; and position 4 is substituted with F and position 2 is substituted with Cl, in 8q-A.

**Figure 3 molecules-25-00895-f003:**
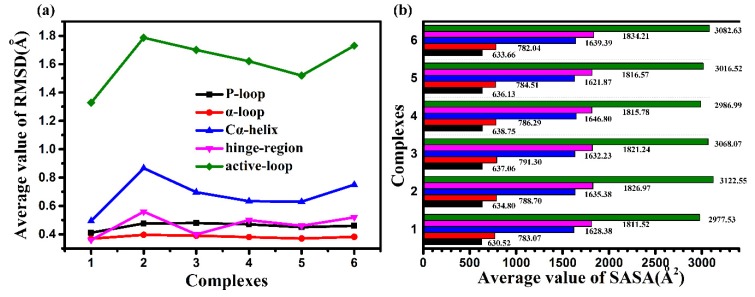
The average value of RMSD (**a**) and solvent-accessible surface area (SASA) (**b**) for the active binding pocket. Complexes 1-6 represent 8r-B-bound, 8r-A-bound, 8p-B-bound, 8p-A-bound, 8q-B-bound, and 8q-A-bound, respectively.

**Figure 4 molecules-25-00895-f004:**
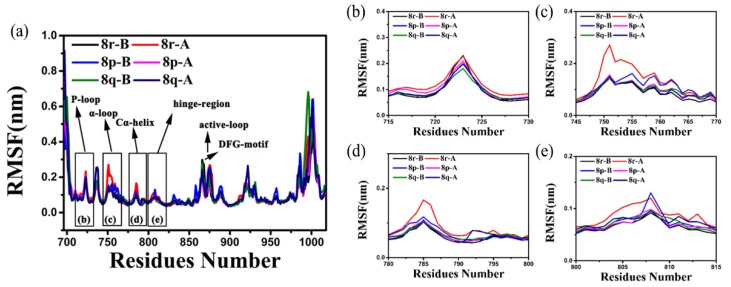
Comparison of root-mean-square fluctuation (RMSF) of protein for EGFR^TM^-8r-B (black line), EGFR^TM^-8r-A (red line), EGFR^TM^-8p-B (blue line), EGFR^TM^-8p-A (magenta line), EGFR^TM^-8q-B (olive line), and EGFR^TM^-8q-A (navy line) illustrated in (**a**). (**b**)The RMSF values of the Cα atoms of Ile715–Leu730 residues. (**c**) The RMSF values of the backbone atoms of Lys745–Asp770 residues. (**d**) The RMSF values of backbone atoms of Ile780–Asp800 residues. (**e**) The RMSF values of the backbone atoms of Asp800–Leu815 residues.

**Figure 5 molecules-25-00895-f005:**
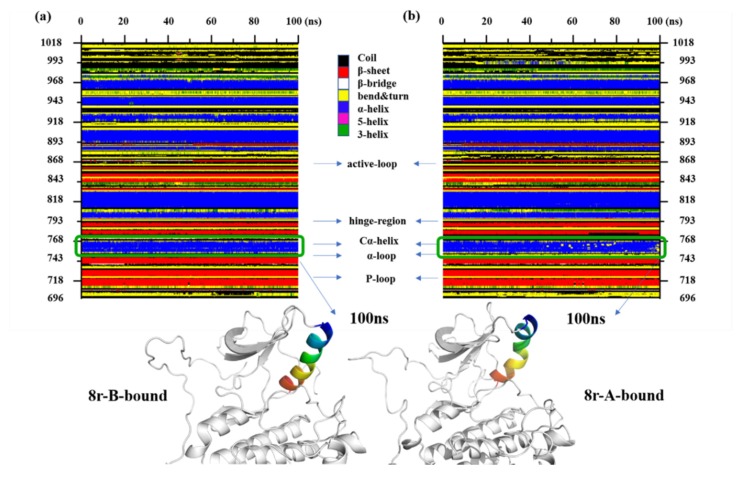
Comparison the differences of secondary structural differences between (**a**) 8r-B-bound, (**b**) 8r-A-bound.

**Figure 6 molecules-25-00895-f006:**
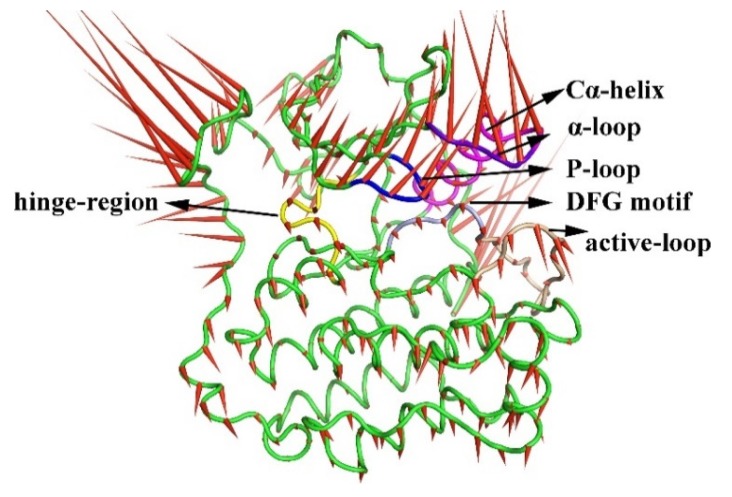
Direction of motion of the Cα skeleton of the apo protein. Green represents the average structure of the protein skeleton, and red arrows represent the direction and trend of skeleton movement.

**Figure 7 molecules-25-00895-f007:**
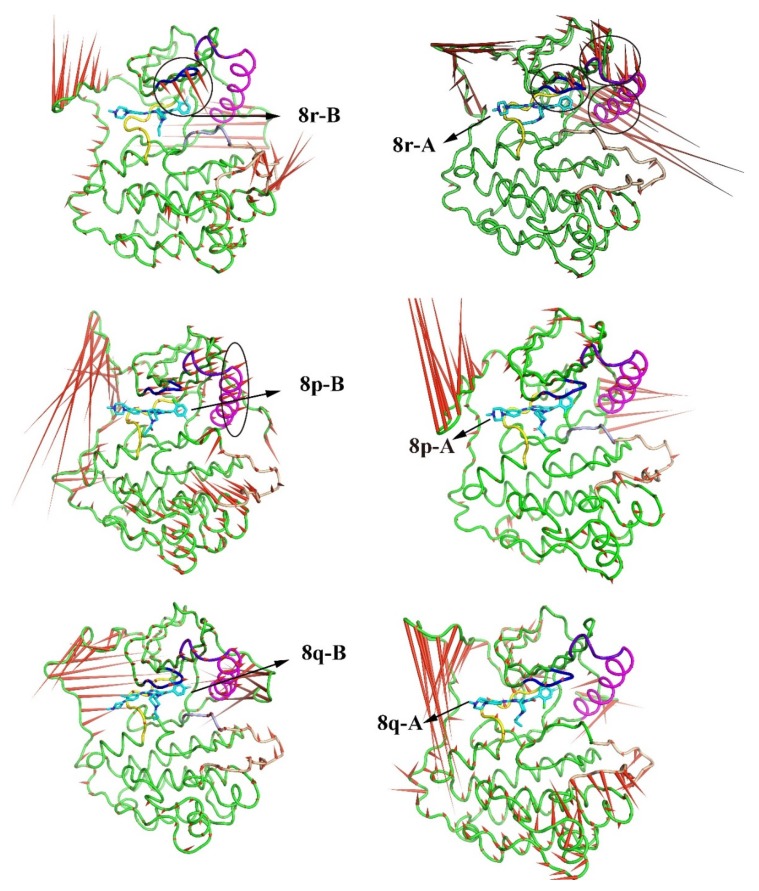
Direction of motion of Cα skeleton of the protein after inhibitor binding. (**a**) Direction of motion of protein after 8r-B binding. (**b**) Direction of motion of protein after 8r-A binding. (**c**) Direction of motion of protein after 8p-B binding. (**d**) Direction of motion of protein after 8p-A binding. (**e**) Direction of motion of protein after 8q-B binding. (**f**) Direction of motion of protein after 8q-A binding. Green represents the average structure of the protein skeleton, and red arrows represent the direction and trend of skeleton movement. The stick structures of the six inhibitors are depicted in Figure 4. The black coil represents the region evidently.

**Figure 8 molecules-25-00895-f008:**
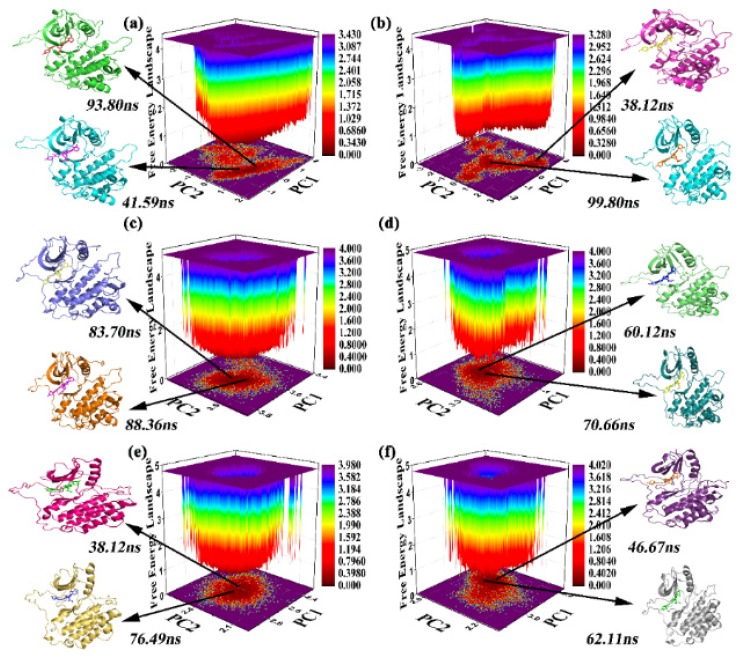
Free energy landscape analysis and representative snapshots extracted from the MD trajectory. Areas in dark blue possess lower energy than other areas. Two representative structures of most populated clusters in (**a**) EGFR^TM^-8r-B, (**b**) EGFR^TM^-8r-A, (**c**) EGFR^TM^-8p-B, (**d**) EGFR^TM^-8p-A, (**e**) EGFR^TM^-8q-B, and (**f**) EGFR^TM^-8q-A complexes.

**Figure 9 molecules-25-00895-f009:**
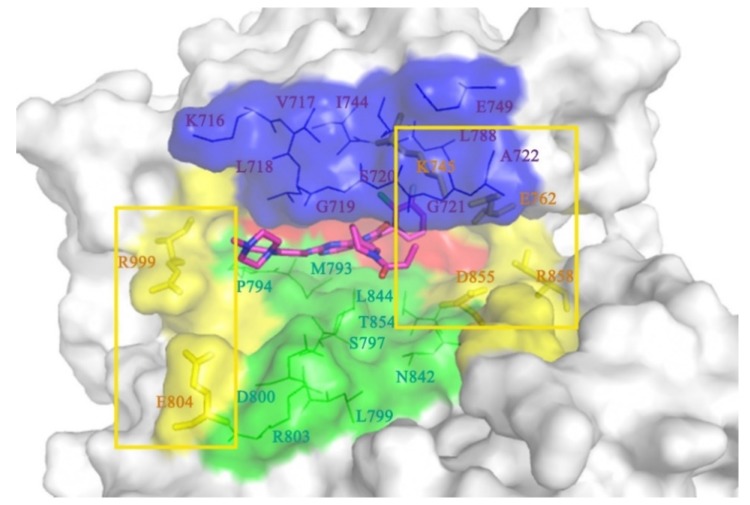
The ‘mouth-like’ pocket of residues around ligands in the surface for surface modality and line modality. K745, E762, D855, and R858 form a system of four-salt bridges marked by a yellow circle on the right side; R999 and E804 form the salt bridge marked by a yellow circle on the left side.

**Figure 10 molecules-25-00895-f010:**
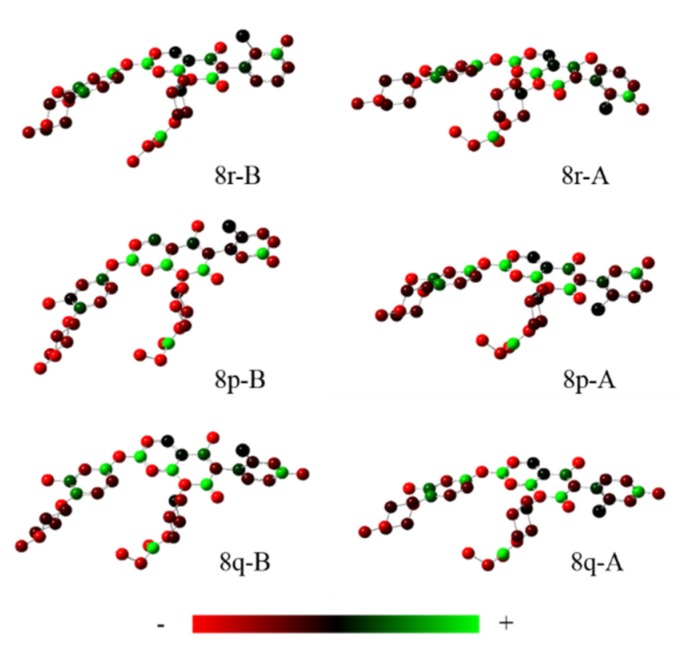
The drawing is based on the charge difference of atoms with different ligands (computed by the Gaussian 09 program); (**a**) 8r-B, (**b**) 8p-B, (**c**) 8q-B, (**d**) 8r-A, (**e**) 8p-A, and (**f**) 8q-A.

**Figure 11 molecules-25-00895-f011:**
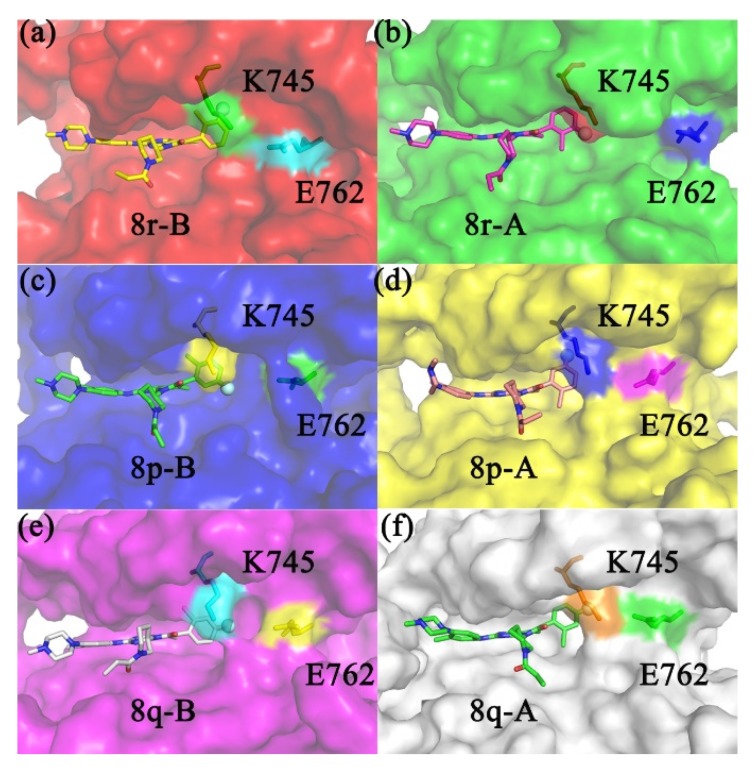
The positional relationship between Lys745, Glu762, and the six inhibitors. (**a**) Location of the inhibitor 8r-B, (**b**) 8r-A, (**c**) 8p-B, (**d**) 8p-A, (**e**) 8q-B, (**f**) 8p-A, in the protein binding pocket. All inhibitors and residues attached to Lys745 and Glu762 are displayed as stick models. The presence of the F atom in the inhibitor is marked by a solid sphere.

**Figure 12 molecules-25-00895-f012:**
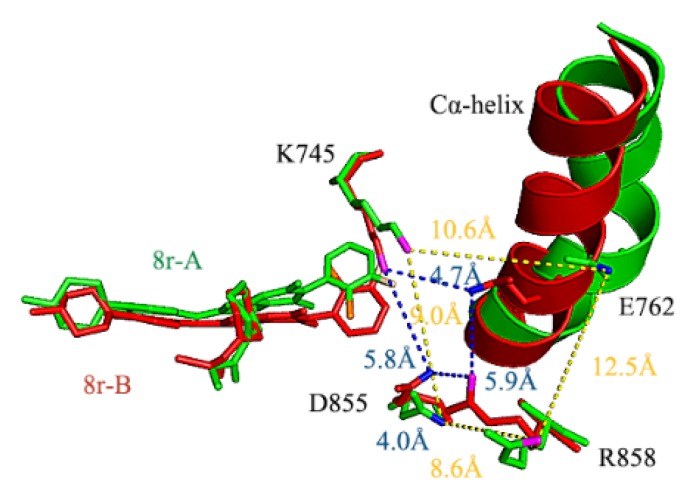
The positional relationship between the inhibitors 8r-B and 8r-A and the Cα-helix. EGFR^TM^-8r-B is marked in red, EGFR^TM^-8r-A is marked in green; residues Lys745, Glu762, Asp855, and Arg858 and the inhibitors are represented by stick figures; the Cα-helix region is represented by a cartoon figure.

**Figure 13 molecules-25-00895-f013:**
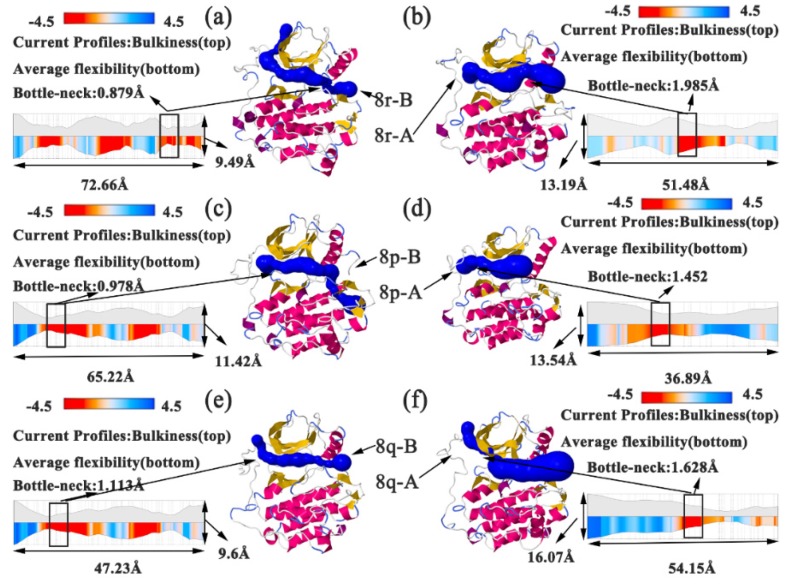
Tunnel profile visualization of EGFR^TM^-8r-B, EGFR^TM^-8r-A, EGFR^TM^-8p-B, EGFR^TM^-8p-A, EGFR^TM^-8q-B, and EGFR^TM^-8q-A complexes, as predicted by Channel Extraction and Visualization (CHEXVIS), and denoted by (**a**–**f**), respectively.

**Figure 14 molecules-25-00895-f014:**
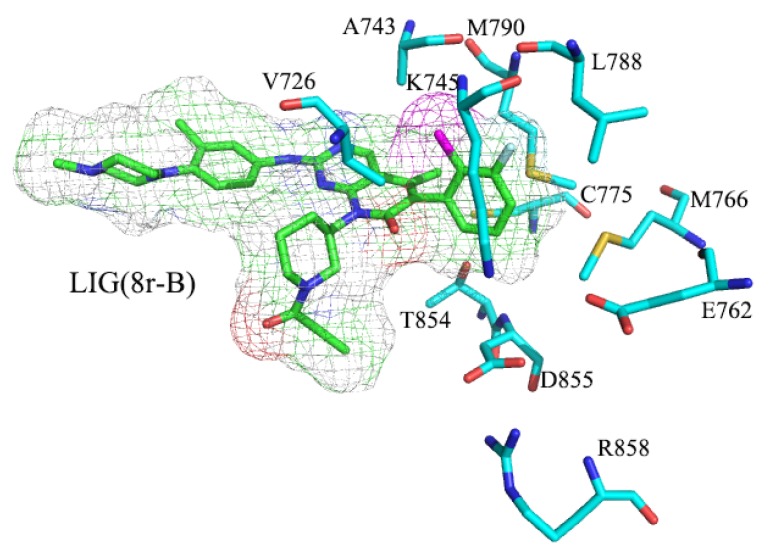
Ligands and amino acids around the benzene ring in the head. The ligands are denoted by the club model and electron density map. The surrounding amino acids are denoted by the club model, and all hydrogen bonds are hidden.

**Table 1 molecules-25-00895-t001:** The main small molecule inhibitors targeted epidermal growth factor receptor (EGFR) for all generations.

Generations of Inhibitors	Name	Research and Development Status	Primary Treatment	Key Achievements	Drawbacks
First	Iressa 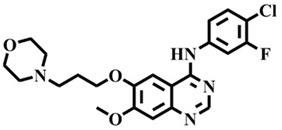	approved in 2003	Deletion in exon 19 (62.2%) and L858R (37.8%) [31,32]	Reversible bind to the mutated substrate pocket, competitive ATP catalytic region of EGFR-TK on binding cell surface	Focused on combining with other therapies [33], easy to occur secondary acquired drug resistance mutations, such as T790M. The incidence of side effects such as rash and diarrhea is high.
	Taeceva 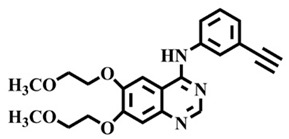	approved in 2004			
	Conmana 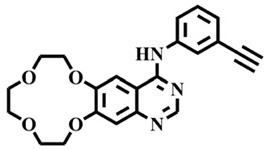	approved in 2011			
Second	Giotrif 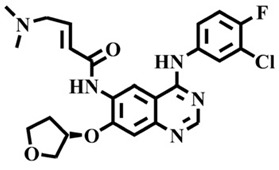	approved in 2013	Approximately 60% of the EGFR gene undergoes a secondary mutation T790M in exon 20	Electrophilic Michael receptor radical group irreversible bind to the mutated substrate pocket with the nucleophilic Cys797, which leads to inhibition of ATP binding	Lack of selectivity for mutant and wild type, the incidence of side effects such as rash and diarrhea, nausea, fatigue.
	Dacomitibib 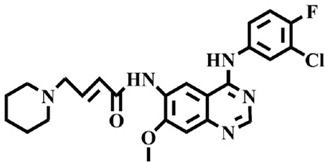	phase Ⅲ			
	Neratinib 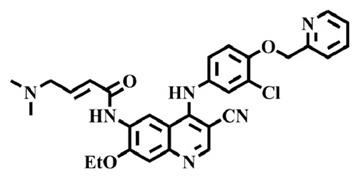	phase Ⅲ			
Third	Rociletinib 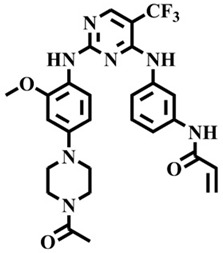	phase I/II	L858R/T790M [34,35,36,37]	The inhibition of mutant protein was higher than that of wild-type protein	Easy to cause hyperglycemia (53%)
	Osimertinib 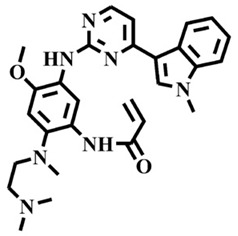	phase I/II			Side effects include diarrhea (47%), nausea (22%), rash, and hemorrhoids (40%)
Fourth	EAI045 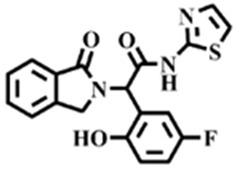	designed in 2016, under clinical test	L858R/T790M/C797S	Allosteric noncompetitive inhibitors	Needs to be combined with EGFR monoclonal antibody
	Brigatinib 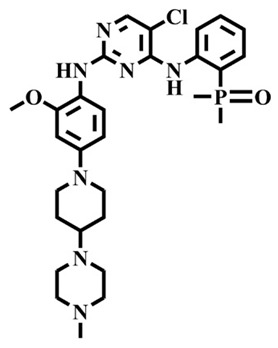	designed in 2017, under clinical test		Belongs to double-target reversible small molecule inhibitor	Dual-target combination therapy, not easy to control

**Table 2 molecules-25-00895-t002:** Docking affinities and root-mean-square deviation (RMSD) of the six inhibitors with the EGFR^TM^.

Complex	EGFR^TM^_8r-B	EGFR^TM^_8r-A	EGFR^TM^_8p-B	EGFR^TM^_8p-A	EGFR^TM^_8q-B	EGFR^TM^_8q-A
RMSD (nm)	0.14	0.29	0.24	0.20	0.21	0.23
Affinity(kcal/mol)	−10.9	−8.0	−10.2	−8.7	−10.7	−8.8

**Table 3 molecules-25-00895-t003:** Hydrogen bond occupancy rates above the molecular dynamics (MD) simulation between Met793 and substrates.

Donor	Acceptor	8r-B	8r-A	8p-B	8p-A	8q-B	8q-A
Met793@N	lig@N4	75.53%	66.53%	75.32%	76.12%	73.53%	72.43%
lig@N11	Met793@O	76.32%	62.44%	72.43%	69.93%	74.23%	68.03%
average hydrogen bond number		2.05	1.99	2.09	1.95	1.89	2.00

**Table 4 molecules-25-00895-t004:** The charge distribution analysis of F and Cl atoms. RESP: Restrained ElectroStatic Potential.

Inhibitors		8r-B	8r-A	8p-B	8p-A	8q-B	8q-A
RESP charge (e)	F	−0.278	−0.282	−0.276	−0.279	−0.273	−0.270
	Cl	−0.072	−0.066	−0.108	−0.102	−0.100	−0.095
Mulliken charge (e)	F	−0.277	−0.277	−0.295	−0.296	−0.293	−0.293
	Cl	+0.014	+0.014	−0.012	−0.012	−0.004	−0.006
NBO charge (e)	F	−0.322	−0.322	−0.331	−0.331	−0.329	−0.329
	Cl	0.021	+0.019	−0.002	−0.004	+0.005	+0.002

**Table 5 molecules-25-00895-t005:** The van der Waals radius of the main atoms wrapped in the benzene ring head.

Atom	C	Cl	F	H
R_vdW_ (Å)	1.72	1.80	1.35	1.10

**Table 6 molecules-25-00895-t006:** Average distance between the Lys745 (NZ), Glu762 (OE2), Lys745 (NZ), and ligand (F) in the whole simulating process.

EGFR^TM^	8r-B	8r-A	8p-B	8p-A	8q-B	8q-A
K745@NZ-L762@OE2	5.52	7.15	6.53	5.75	5.85	5.93
K745@NZ-ligand@F	4.20	4.63	5.03	5.20	4.91	5.21

**Table 7 molecules-25-00895-t007:** The Molecular mechanics Poisson-Boltzmann surface area (MM-PBSA) results. The unit of measurement value is kJ·mol^−1^.

Energy Term	8r-B	8r-A	8p-B	8p-A	8q-B	8q-A
IC_50_ (nM)	27.5 ± 11.6	>1000	37.1 ± 1.2	207.0 ± 135.0	88.6 ± 13.3	224.1 ± 6.7
ΔG_vdW_	−285.88 ± 3.47	−260.23 ± 1.69	−279.78 ± 0.96	−261.40 ± 1.96	−258.41 ± 1.72	−254.67 ± 1.59
ΔG_elec_	−107.63 ± 1.70	−99.30 ± 2.93	−93.06 ± 1.28	−102.59 ± 2.85	−85.20 ± 2.55	−91.48 ± 2.10
ΔG_PB_	177.89 ± 1.94	180.51 ± 3.31	177.06 ± 1.18	177.34 ± 2.88	151.13 ± 2.05	161.00 ± 2.16
ΔG_np_	−26.26 ± 0.31	−24.99 ± 0.19	−25.80 ± 0.08	−24.57 ± 0.14	−24.40 ± 0.14	−24.12 ± 0.16
ΔG_polar_ ^a^	70.26 ± 3.64	81.21 ± 6.24	84.00 ± 2.46	74.75 ± 5.73	65.94 ± 4.60	69.52 ± 4.26
ΔG_nonpolar_ ^b^	−312.14 ± 3.78	−285.23 ± 1.88	−305.58 ± 1.04	−285.97 ± 2.1	−282.81 ± 1.86	−278.79 ± 1.75
ΔG_bind_ ^c^	−241.80 ± 3.44	−203.63 ± 3.57	−221.46 ± 1.41	−210.81 ± 2.56	−216.94 ± 2.42	−208.93 ± 2.75

(^a^ ΔG_polar_ = ΔG_elec_ + ΔG_PB_, ^b^ ΔG _nonpolar_= ΔG_vdW_ + ΔG_np_, ^c^ ΔG_bind_ = ΔG_polar_+ ΔG_nonpolar_).

**Table 8 molecules-25-00895-t008:** Binding free energy of decomposition. The unit of measurement is kcal/mol.

Energy Term	8r-B	8r-A	8p-B	8p-A	8q-B	8q-A
IC_50_ (nM)	27.5 ± 11.6	>1000	37.1 ± 1.2	207.0 ± 135.0	88.6 ± 13.3	224.1 ± 6.7
Val726	−10.27 ^a^	−7.90	−9.54	−8.21	−9.61	−7.98
	1.48 ^b^	1.47	1.41	1.68	1.52	1.42
	−9.61 ^c^	−7.2	−9.02	−7.39	−8.93	−7.44
Ala743	−5.77	−7.63	−6.59	−5.06	−6.64	−6.41
	2.35	3.65	2.50	3.06	2.31	3.02
	−3.72	−4.33	−4.43	−2.41	−4.66	−3.74
Lys745	−27.25	−22.15	−23.20	−22.34	−20.19	−19.68
	26.15	28.72	23.27	32.03	21.12	20.67
	−2.05	5.83	−0.88	9.07	0.34	0.36
Glu762	−2.19	6.68	3.40	−2.18	3.05	2.28
	6.45	3.57	1.32	2.77	0.63	−0.44
	4.22	10.27	4.70	0.56	3.66	1.82
Met766	−3.41	−1.79	−3.83	−2.98	−2.43	−2.31
	1.97	1.28	1.34	1.81	1.02	1.56
	−1.66	−0.73	−2.72	−1.38	−1.64	−0.95
Cys775	−1.92	−2.12	−2.25	−2.42	−1.77	−2.09
	0.60	0.91	0.81	1.02	0.61	0.82
	−1.40	−1.26	−1.47	−1.46	−1.17	−1.32
Leu788	−1.47	−4.96	−1.47	−1.86	−2.20	−5.04
	0.80	1.40	0.05	0.90	0.72	1.33
	−0.77	−3.65	−1.50	−1.07	−1.58	−3.80
Met790	−6.45	−5.96	−7.32	−6.18	−6.53	−5.78
	1.15	1.68	0.84	1.97	1.23	1.83
	−5.96	−4.88	−7.08	−4.92	−5.94	−4.64
Thr854	−1.92	−4.86	−3.41	−3.51	−3.77	−4.28
	2.32	3.74	5.44	3.93	0.43	4.45
	0.05	−1.52	1.58	0.13	−3.78	−0.15
Asp855	−0.73	6.02	6.52	−0.83	−3.22	−0.10
	4.55	8.48	1.84	4.56	8.83	3.38
	3.55	14.00	8.21	3.33	5.18	2.86
Arg858	−0.79	−4.02	−3.83	−1.38	−0.97	2.26
	−0.65	0.23	−0.10	−0.34	−0.73	0.04
	−1.44	−3.78	−3.94	−1.74	−1.70	−2.22

(^a^ ΔG_polar_ = ΔG_elec_ + ΔG_PB_, ^b^ ΔG_nonpolar_ = ΔG_vdW_ + ΔG_np_, ^c^ ΔG_bind_ = ΔG_polar_+ ΔG_nonpolar_).

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
