# Peer review of "How Different Substitution Positions of F, Cl Atoms in Benzene Ring of 5-Methylpyrimidine Pyridine Derivatives Affect the Inhibition Ability of EGFRL858R/T790M/C797S Inhibitors: A Molecular Dynamics Simulation Study"

_molecules, 2020, doi:10.3390/molecules25040895_

Round 1
Reviewer 1 Report
The manuscript describes an interesting topic related to medicinal chemistry. However, some points need improvement:
there are several typos and language erros. all figures have bad quality. page 1, lines 38-42: the sentence is confuse. Chart 1: all structures have bad quality. correct the termo "IC50" along the text (the number 50 should be subscript). section 2.1: this topic shoud include data the literature for the similar systems to corroborate the discussion on the main interactions. page 5, line 122: correct Lus745. Table 1: affinity or value from scoring function? Is it correct to consider these values as affinity? page 6, lines 149-150: something is missing from the following sentence: "It is now a commonly accepted concept that the active site of the enzyme located in the cavity of the hydrophobic environment." Figure 3: make it clear who is the 8r-B in figure a. Figure 6: is "pure protein" a suitable term? page 11, line 262: "the cluster analysis method was obtained a series of stable..." - revise writing. page 12, lines 292-298: is there evidence in the literature on similar systems that corroborate these findings? Table 3: what kind of method was used to analyse the charges? RESP should be the main method to be considered in the analyses. page 17, line 459: "interactions (ΔG vdW) of the 8r-B complex are greater than those of the other inhibitors." Greater??? Shouldn't these values be lower? Revise this point in this section. Table 6: put the IC50 value in the table. page 18, line 471: large? page 20, line 543: redocking test showed a relatively high value and the figure from redocking indicates a region with a significant deviation. Some docking parameters could be modified to improve the quality of the results.Author Response
Response to Reviewer 1 Comments
Point 1: There are several typos and language errors. All figures have bad quality. Chart 1: all structures have bad quality. Correct the term "IC50" along the text (the number 50 should be subscript).
Response 1: We’d like to show our heartfelt thanks to you for your kindly and helpful comments. As you suggested that the manuscript has been carefully revised considering about the language and grammar problems. And we have made some adjustments to improve the quality of Chart 1 figures and have rearranged the pictures. The term "IC50" along the text has been rectified.
Point 2: Page 1, lines 38-42: the sentence is confused.
Response 2: Thank you for your careful correction. the sentence has been rewritten as “When some protein kinases over expression or abnormal expression in cancer cells leads to abnormal signal transduction, which is involved in many physiological processes such as cell proliferation, differentiation, metabolism, apoptosis, etc. The pathogenesis of non-small cell lung cancer (NSCLC) is closely related to the mutant kinase activity in epidermal growth factor receptor (EGFR), abnormal signal transduction is closely related to the occurrence of cancer, it can accelerate cellular apoptosis, antagonize angiogenesis, and inhibit tumor metastasis and tumor growth.”
Point 3: Section 2.1: this topic should include data of the literature for the similar systems to corroborate the discussion on the main interactions.
Response 3: Thank you for your helpful comment. There are Page 5, lines 112-113 sentence has been rewritten as “All six inhibitors belong to reversible inhibitors which located in the cavity of the hydrophobic environment, and the location of these competitive inhibitors is the reported ATP binding site”, and added six similar systems follow the sentence, there are
Park, H.; Jung, H.-Y.; Mah, S.; Hong, S., Discovery of EGF Receptor Inhibitors That Are Selective for the d746-750/T790M/C797S Mutant through Structure-Based de Novo Design. Angewandte Chemie International Edition 2017, 56 (26), 7634-7638. Yu, L.; Huang, M.; Xu, T.; Tong, L.; Yan, X. E.; Zhang, Z.; Xu, Y.; Yun, C.; Xie, H.; Ding, K.; Lu, X., A structure-guided optimization of pyrido[2,3-d]pyrimidin-7-ones as selective inhibitors of EGFR(L858R/T790M) mutant with improved pharmacokinetic properties. Eur J Med Chem 2017, 126, 1107-1117. To, C.; Jang, J.; Chen, T.; Park, E.; Mushajiang, M.; De Clercq, D. J. H.; Xu, M.; Wang, S.; Cameron, M. D.; Heppner, D. E.; Shin, B. H.; Gero, T. W.; Yang, A.; Dahlberg, S. E.; Wong, K. K.; Eck, M. J.; Gray, N. S.; Janne, P. A., Single and Dual Targeting of Mutant EGFR with an Allosteric Inhibitor. Cancer Discov 2019, 9 (7), 926-943. Wan, S.; Yan, R.; Jiang, Y.; Li, Z.; Zhang, J.; Wu, X., Insight into binding mechanisms of EGFR allosteric inhibitors using molecular dynamics simulations and free energy calculations. J Biomol Struct Dyn 2019, 37 (16), 4384-4394. Akher, F. B.; Farrokhzadeh, A.; Soliman, M. E. S., Noteworthy effect of slight variation in aliphatic chain length of trisubstituted imidazole inhibitors against epidermal growth factor receptor L858R/T790M/C797S mutant in cancer therapy. Chem Biol Drug Des 2019, 93 (5), 798-810. Li, Q.; Zhang, T.; Li, S.; Tong, L.; Li, J.; Su, Z.; Feng, F.; Sun, D.; Tong, Y.; Wang, X.; Zhao, Z.; Zhu, L.; Ding, J.; Li, H.; Xie, H.; Xu, Y., Discovery of Potent and Noncovalent Reversible EGFR Kinase Inhibitors of EGFR(L858R/T790M/C797S). ACS Med Chem Lett 2019, 10 (6), 869-873.
Point 4: Page 5, line 122: correct Lus745. Table 1: affinity or value from scoring function? Is it correct to consider these values as affinity?
Response 4: Thanks for your question. We are sorry for the mistake, now the wrong wording has corrected as “Lys745”. The scoring function was used the default scoring function of Autodock Vina software, For the estimation of ligand-receptor affinity, Vina uses an empirical scoring function which is inspired by the X-score function. Generally, the higher the affinity between the inhibitor and the protein, the more stable the binding, and the stronger the inhibition effect. In addition, the relevant references are added at the Page 20, lines 531. Therefore, the accuracy of our calculation results, the method of evaluation and the expression of writing can be guaranteed.
Wang, R.; Lai, L.; Wang, S., Further development and validation of empirical scoring functions for structure-based binding affinity prediction. Journal of computer-aided molecular design 2002, 16 (1), 11-26. Trott, O.; Olson, A. J., AutoDock Vina: improving the speed and accuracy of docking with a new scoring function, efficient optimization, and multithreading. Journal of computational chemistry 2010, 31 (2), 455-461. Quiroga, R.; Villarreal, M. A., Vinardo: A scoring function based on autodock vina improves scoring, docking, and virtual screening. PloS one 2016, 11 (5), e0155183. Trott, O.; Olson, A. J., AutoDock Vina: improving the speed and accuracy of docking with a new scoring function, efficient optimization, and multithreading. J Comput Chem 2010, 31 (2), 455-61.
Point 5: Page 6, lines 149-150: something is missing from the following sentence: "It is now a commonly accepted concept that the active site of the enzyme located in the cavity of the hydrophobic environment.
Response 5: Thank you for your careful correction. the sentence has been rewritten as “It is now a commonly accepted concept that the active site of the enzyme located in the hydrophobic cavity of protein.”
Point 6: Figure 3: make it clear who is the 8r-B in Figure a.
Response 6: Very grateful to your comments, which is helpful for improving our manuscript. We have reorganized the language and corrected the name of each complex at the bottom of the paragraph. Please see the revised version. In the Page 6, Line 159-160 are listed as follow: “Figure 3. The average value of RMSD (a) and SASA (b) for the active binding pocket. Complexes 1-6 represent 8r-B-bound, 8r-A-bound, 8p-B-bound, 8p-A-bound, 8q-B-bound, 8q-A-bound, respectively.”
Point 7: Figure 6: is "pure protein" a suitable term?
Response 7: Thank you for your helpful correction. We went through the other articles carefully and revised the term of "pure protein" to "apo protein". The reference articles are listed as follow:
Scapin G, Patel S, Patel V, et al. The structure of apo protein‐tyrosine phosphatase 1B C215S mutant: More than just an S→ O change[J]. Protein Science, 2001, 10(8): 1596-1605.
Pedersen A K, Peters G H, Møller K B, et al. Water-molecule network and active-site flexibility of apo protein tyrosine phosphatase 1B[J]. Acta Crystallographica Section D: Biological Crystallography, 2004, 60(9): 1527-1534.
Morita M, Nakamura S, Shimizu K. Highly accurate method for ligand‐binding site prediction in unbound state (apo) protein structures[J]. Proteins: Structure, Function, and Bioinformatics, 2008, 73(2): 468-479.
Babor M, Gerzon S, Raveh B, et al. Prediction of transition metal‐binding sites from apo protein structures[J]. Proteins: Structure, Function, and Bioinformatics, 2008, 70(1): 208-217.
Point 8: Page 11, line 262: "the cluster analysis method was obtained a series of stable..." - revise writing.
Response 8: Thank you for your careful correction. the sentence has been rewritten as “Then, the cluster analysis method was applied to obtain a series of stable conformations from the equilibrium trajectory, shown as S5(a-f).”
Point 9: Page 12, lines 292-298: is there evidence in the literature or similar systems that corroborate these findings?
Response 9: Thank you for your helpful comment. There are some similar systems added to the article at Page 12, lines 298, there are
Dixit, A.; Verkhivker, G. M., Hierarchical modeling of activation mechanisms in the ABL and EGFR kinase domains: thermodynamic and mechanistic catalysts of kinase activation by cancer mutations. PLoS Comput Biol 2009, 5 (8), e1000487. Ju, Y.; Wu, J.; Yuan, X.; Zhao, L.; Zhang, G.; Li, C.; Qiao, R., Design and Evaluation of Potent EGFR Inhibitors through the Incorporation of Macrocyclic Polyamine Moieties into the 4-Anilinoquinazoline Scaffold. J Med Chem 2018, 61 (24), 11372-11383.
Point 10: Table 3: what kind of method was used to analysis the charges? RESP should be the main method to be considered in the analyses.”
Response 10: We’d like to show our thanks to your helpful comments. Mulliken method was used to analysis the charges, but it is not appropriate, the Mulliken change combine with RESP charge and NBO change have be added to the charge distribution analysis. And RESP is the main method to be considered in the analyses. Please see the revised version. In the Page 13, Line 323-327 are listed as follow: “Interestingly, the negative charge of F atom is strong while the negative charge of Cl atom is not high and approximately 0, that is, the Cl-atom carries no charge in all inhibitors. Moreover, the charge difference of atoms with different ligands in RESP methods showed the F-atom had the highest negative charge and the difference of charge between the six inhibitors is insignificant (as represented in Figure 10).”.
Point 11: Page 17, line 459: "interactions (ΔG vdW) of the 8r-B complex are greater than those of the other inhibitors." Greater??? Shouldn't these values be lower? Page 18, line 471: large? Revise this point in this section.
Response 11: Thank you for your careful correction. We are sorry for the mistake, now the wrong wording has corrected as “the 8r-B complex are lower than those of the other inhibitors” and “Firstly, the residues around the F-atom possess relatively lowest values of binding free energies, though they change when the F-atom is located at a different site.”
Point 12: Table 6: put the IC50 value in the table.
Response 12: Thank your comment, according to your suggestion, the IC50 value has been put in Table 6.
Point 13: Page 20, line 543: redocking test showed a relatively high value and the figure from redocking indicates a region with a significant deviation. Some docking parameters could be modified to improve the quality of the results.
Response 13: Thank you for your comment, it is very useful to us. According to your opinion, we have some adjustment docking parameter in the method section and have rearrange Figure S9. The original inhibitor CO-1686 in 5XDK.pdb was selected as the contract for redocking analysis. The size of the docking box chosen was 12×14×18 points in the x, y, and z-axis directions with a gird spacing of 1 Å for CO-1686, the center of the grid was regarded as the center of mass and the exhaustiveness was 20. The pose of the ligand CO-1686 obtained from the docking resembled the one determined by X-ray crystallography, and the RMSD (root-mean-square deviation) of heavy atoms in our docking calculation was only 2.001 Å, demonstrating the high reliability of our docking scenario. The resulting was used here as a reference for method validation. The adjustment docking parameter is in the Page 20, Line 548-554, listed as follow:
“The double hydrogen bond formed by Met790 and inhibitor was well reproduced in the result in Figure S9 (a), the difference between the crystal ligand and the redocking model lied in the tail end of the inhibitor piperazine ring which was exposed to the outside of the protein and entered the polar solvent, which was the reason for the large fluctuation. In the Figure S9 (b-c), the residues which involved van der Waals interactions and electrostatic interactions are the same, except of the ethanoylpiperazin of ligand, other atoms were reasonably presented high coincidence.”

Reviewer 2 Report
The paper by Jingwen and coworkers studies at the atomic level the effect of halogen substitution on a set of fourth generation inhibitors of mutated EGFR. Using docking, molecular dynamics, PCA analysis, Free energy calculations and other tools, the authors analyse the effect of the position of F and Cl atoms on a phenyl ring in otherwise identical molecules.
The authors succeed in identifying and justifying the structural reasons for the different activity of the different molecules they consider and very well correlate their results with the available experimental data.
On the whole, the work is well performed, only minor revisions are required prior to publication. In particular, a linguistic revision is needed, in particular of the parts written in red in the paper, which seems to have been written not as accurately as the rest of the paper.
Moreover:
In Fig. 3 panel b, it is very hard to notice any difference in the SASA data. It would be necessary to assess if such data are actually significant and to give some statistics.
RMSF data should be analysed more deeply to test id they are significant, for example as shown in Mitra and Sept, Biophysical Journal, 2008, 95, 3252.
In Fig. 10 Caption Gaussian 9.0 should be Gaussian09 (as correctly reported in the materials and methods section).In the materials and methods section, the paragraph "Initial system preparation" is a bit unclear. On line 516 it is said that a molecular dynamics has been performed. Then on line 518 the authors talk about "optimized structure". In this phase the system underwent MD or geometry optimization, or both? The selected initial structure was the final structure of the MD , further optimised or something else?
In the "Molecular Dynamics (MD) Simulation" section, I guess that the GAFF force field has been used for the ligands. All force fields and water model employed (GAFF, AMBER99SB-ILDN, SPC water model) should be properly referenced with the relevant papers citation.
Author Response
Point 1: The paper by Jingwen and coworkers studies at the atomic level the effect of halogen substitution on a set of fourth generation inhibitors of mutated EGFR. Using docking, molecular dynamics, PCA analysis, Free energy calculations and other tools, the authors analysis the effect of the position of F and Cl atoms on a phenyl ring in otherwise identical molecules. The authors succeed in identifying and justifying the structural reasons for the different activity of the different molecules they consider and very well correlate their results with the available experimental data. On the whole, the work is well performed, only minor revisions are required prior to publication. In particular, a linguistic revision is needed, in particular of the parts written in red in the paper, which seems to have been written not as accurately as the rest of the paper.
Response 1: We’d like to show our heartfelt thanks to the referees for his kindly and
helpful comments. Your affirmation is a great inspiration to us.
Point 2: In Fig. 3 panel b, it is very hard to notice any difference in the SASA data. It would be necessary to assess if such data are actually significant and to give some statistics.
Response 2: Very grateful to your comments, which is helpful for improving our manuscript. Both the RMSD and SASA date could showed that 8r-B-bound complex provide a decent stable and hydrophobic environment for the substrate binding. We have reorganized the Figure 3 include the value of every active site to make it clear.
Point 3: RMSF data should be analyzed more deeply to test if they are significant, for example as shown in Mitra and Sept, Biophysical Journal, 2008, 95, 3252.
Response 3: We have studied your comments carefully and truly grateful to your thoughtful suggestions, this article is very helpful for us which has been the relevant citation added in article. The revised version listed as follow:
The Page 6, lines 162-165 sentence has been rewritten as “RMSF is one of the best methods of comparing dynamics and profiles of all residues of the protein Cα are represented in Figure 4 (a), parts of the protein that highly flexible will have a large RMSF value while portions that are constrained will result in a low RMSF”,
Page 6, lines 162-165 sentence has been rewritten as “It is usual for the unconstrained and outer-sphere ending loop. Except for the residues from Leu979 to Ile1018, like the P-loop (Ser720 to Glu724), α-loop (Arg748 to Ser752), Cα-helix (Pro753 to Ser768), hinge-region (Gln791 to Leu798), and the active-loop (DFG-motif, Thr854 to Pro877), which comprise the inhibitors binding functional regions, exhibited relatively flexible and significant in different complexes.”
Page 6, lines 176-179 sentence has been rewritten as “The P-loop, α-loop, Cα-helix, and active-loop are all present at the phenyl terminal of the substrate, in the form of a benzene ring with halogen substitutions, the phenyl rings could show a fair degree of mobility and explore a relative wide range of conformations when the four regions exhibited relatively flexible.”
Point 4: In Fig. 10 Caption Gaussian 9.0 should be Gaussian 09 (as correctly reported in the materials and methods section).
Response 4: Thank you for your careful correction. We are sorry for the mistake, the correct term - “Gaussian 09” has been corrected in the manuscript.
Point 5: In the materials and methods section, the paragraph "Initial system preparation" is a bit unclear. On line 516 it is said that a molecular dynamic has been performed. Then on line 518 the authors talk about "optimized structure". In this phase the system underwent MD or geometry optimization, or both? The selected initial structure was the final structure of the MD, further optimized or something else?
Response 5: We’d like to show our heartfelt thanks to your helpful comments. The selected initial structure was the final structure of the MD, we are sorry for the confused sentence and has corrected the manuscript in the Page 20, Line 527-528, the revised version listed as follow:
“therefore, the final structure of the apo protein molecular dynamics simulation could be selected as the initial model for next simulation.”
Point 6: In the "Molecular Dynamics (MD) Simulation" section, I guess that the GAFF force field has been used for the ligands. All force fields and water model employed (GAFF, AMBER99SB-ILDN, SPC water model) should be properly referenced with the relevant papers’ citation.
Response 6: Thank you for your helpful correction. The properly referenced with the relevant paper’ citation for “GAFF, AMBER99SB-ILDN, SPC water model” have been added in article. The relevant version listed as follow:
For GAFF: 69. Dickson, C. J.; Rosso, L.; Betz, R. M.; Walker, R. C.; Gould, I. R., GAFFlipid: a General Amber Force Field for the accurate molecular dynamics simulation of phospholipid. Soft Matter 2012, 8 (37), 9617-9627.
For AMBER99SB-ILDN:67. Lindorff‐Larsen, K.; Piana, S.; Palmo, K.; Maragakis, P.; Klepeis, J. L.; Dror, R. O.; Shaw, D. E., Improved side‐chain torsion potentials for the Amber ff99SB protein force field. Proteins: Structure, Function, and Bioinformatics 2010, 78 (8), 1950-1958.
For SPC: 70. Hess, B.; van der Vegt, N. F., Hydration thermodynamic properties of amino acid analogues: a systematic comparison of biomolecular force fields and water models. The journal of physical chemistry B 2006, 110 (35), 17616-17626.

Round 2
Reviewer 1 Report
The revised version of the manuscript is suitable for publication.
This manuscript is a resubmission of an earlier submission. The following is a list of the peer review reports and author responses from that submission.
Round 1
Reviewer 1 Report
The manuscript describes an interesting study using computational tools in order to understand the inhibition mechanism of EGFR. However, some points need improvements:
1) check typos and language.
2) improve the quality of all figures.
3) check the numerical sequence of figures in the main text and Supplementary Material (sometimes, it is incorrect).
4) page 4, lines 99-101: I suggest the authors discuss the interactions with the main residues based on results from the literature.
5) Table 1: is the affinity values obtained from docking reliable? What scoring function was used?
6) page 10, line 248: the main weakness of this analysis refers to the method employed to calculate of charges. Mulliken is not appropriate, the authors should use another calculation methodology, for example, ESP.
7) Tables 3 and 4 can be joined.
8) page 17, line 390: what was the criterion to choose the PDB structure?
9) Was a docking protocol to validate the procedure used? For example, redocking?
10) The section "Conclusions" should be rewriten emphasizing the main results obtained in all analyses performed in the study.
Reviewer 2 Report
In the Manuscript titled “How different substitution positions of F, Cl atoms in benzene ring of 5-methylpyrimidine pyridine derivatives affect the inhibition ability of EGFRL858R/T790M/C797S Inhibitors: A Molecular Dynamics Simulation Study” authors studied “The inhibitory mechanism and structure-activity relationship of six ligand towards EGFR. The authors have used computational modeling and molecular dynamics simulation methods to understand the inhibitory activity is highly appreciable however the manuscript has serious methodological errors, so it is not recommended for publication in the current form. A major revision is required at this stage.
The aim of the study is not clearly explained and the organization of the manuscript is not clear, need to be revised completely with appropriate conceptual flow.
Points to consider:
Did the authors generate the mutated EGFR, if yes why SWISS_MODEL was used? How was the model optimized and what are the parameters used? Figure S1 why there is so much difference in the native and optimized model? What is the percentage of change or similarity? Why 1Å grid spacing is used in molecular docking, justify? In page 415 the grid details. …Figure S2…. But figure S2 is RMSD … In MD method why authors used AMBER99SB_ILDN force field explain since the authors used AMBER14 tools why the recent force field AMBER14SB was not used In MD methods … to neutralize the system several sodium ions were randomly added to the simulation box … is it correct? Why the authors always use C-alpha RMSD why not use all backbone atoms. How much the secondary structural characteristics changes during the simulation? Why the docking results of only Autodock VINA are reported? What is the maximum number of GA runs used? If it is 20 then why the authors claim top 20 solutions were considered? Docking affinities in table 1 is similar for almost three molecules, so how was the best molecule selected? The convergence of sampling is so common now days in MD simulation there is no need for the authors to show this as a main figure instead they can move this to supporting information and it will be good if the trajectory level interactions like inter and intra molecular hydrogen bonding are shown. The porcupine plot for the PCA is highly appreciable but is there any information on the clustering of the sampling? How the free energy landscape funnel is constructed no description is given – how the representative clusters were showed? What does the funnel convey from the over all result?Minor points
Introduction needs to be rewritten with appropriate figures and make the contents clear for the readers for example give flow chart or table for all generations of inhibitors with molecule figure and key achievements and the drawbacks if any etc.,
Follow the protocols of GROMOACS-4.5.2 and correct the method section appropriately. For example …module was used to load the molecule and …. Module was used to add sodium ions ……
In page 17 l# Optimized structures of EGFR and the ligand molecules. ……were employed for (needs to be written appropriately…)
Quote appropriate reference under the method section, for eg. the key contributing paper to the field or the paper related to the methodology development of the package that authors used.